https://doi.org/10.1038/s41467-020-14682-6　　**OPEN**

# *MYCN* amplification and *ATRX* mutations are incompatible in neuroblastoma

Maged Zeineldin[1,18], Sara Federico[2,18], Xiang Chen [3,4,18], Yiping Fan[3], Beisi Xu [3], Elizabeth Stewart[2], Xin Zhou[3], Jongrye Jeon[1], Lyra Griffiths[1], Rosa Nguyen[1], Jackie Norrie[1], John Easton[3], Heather Mulder [3], Donald Yergeau[3], Yanling Liu[3], Jianrong Wu[5], Collin Van Ryn[6], Arlene Naranjo[6], Michael D. Hogarty[7], Marcin M. Kamiński [8], Marc Valentine[9], Shondra M. Pruett-Miller [10], Alberto Pappo[2], Jinghui Zhang [3], Michael R. Clay [11], Armita Bahrami[11], Peter Vogel [11], Seungjae Lee[11], Anang Shelat [12], Jay F. Sarthy[13], Michael P. Meers[13], Rani E. George[14], Elaine R. Mardis [15], Richard K. Wilson[15], Steven Henikoff[13,16], James R. Downing[11] & Michael A. Dyer[1,4,16,17]*

Aggressive cancers often have activating mutations in growth-controlling oncogenes and inactivating mutations in tumor-suppressor genes. In neuroblastoma, amplification of the *MYCN* oncogene and inactivation of the *ATRX* tumor-suppressor gene correlate with high-risk disease and poor prognosis. Here we show that *ATRX* mutations and *MYCN* amplification are mutually exclusive across all ages and stages in neuroblastoma. Using human cell lines and mouse models, we found that elevated *MYCN* expression and *ATRX* mutations are incompatible. Elevated MYCN levels promote metabolic reprogramming, mitochondrial dysfunction, reactive-oxygen species generation, and DNA-replicative stress. The combination of replicative stress caused by defects in the ATRX–histone chaperone complex, and that induced by MYCN-mediated metabolic reprogramming, leads to synthetic lethality. Therefore, *ATRX* and *MYCN* represent an unusual example, where inactivation of a tumor-suppressor gene and activation of an oncogene are incompatible. This synthetic lethality may eventually be exploited to improve outcomes for patients with high-risk neuroblastoma.

[1] Department of Developmental Neurobiology, St. Jude Children's Research Hospital, Memphis, TN 38105, USA. [2] Department of Oncology, St. Jude Children's Research Hospital, Memphis, TN 38105, USA. [3] Department of Computational Biology, St. Jude Children's Research Hospital, Memphis, TN 38105, USA. [4] St. Jude Children's Research Hospital—Washington University Pediatric Cancer Genome Project, St. Louis, MO, USA. [5] Department of Biostatistics, St. Jude Children's Research Hospital, Memphis, TN 38105, USA. [6] Children's Oncology Group Statistics and Data Center, Department of Biostatistics, University of Florida, Gainesville, FIL 32607, USA. [7] Division of Oncology, Children's Hospital of Philadelphia, Philadelphia, PA 19104, USA. [8] Department of Immunology, St. Jude Children's Research Hospital, Memphis, TN 38105, USA. [9] Cytogenetics Shared Resource, St. Jude Children's Research Hospital, Memphis, TN 38105, USA. [10] Center for Advanced Genome Engineering, St. Jude Children's Research Hospital, Memphis, TN 38105, USA. [11] Department of Pathology, St. Jude Children's Research Hospital, Memphis, TN 38105, USA. [12] Department of Chemical Biology and Therapeutics St. Jude Children's Research Hospital, Memphis, TN 38105, USA. [13] Basic Science Division, Fred Hutchinson Cancer Research Center, Seattle, WA 98109, USA. [14] Department of Hematology/Oncology, Dana Farber Cancer Institute, Boston, MA 02215, USA. [15] The Institute for Genomic Medicine, Nationwide Children's Hospital, Columbus, OH 43205, USA. [16] Howard Hughes Medical Institute, Chevy Chase, MD 20815, USA. [17] Department of Ophthalmology, University of Tennessee Health Science Center, Memphis, TN 38163, USA. [18]These authors contributed equally: Maged Zeineldin, Sara Federico, Xiang Chen. *email: michael.dyer@stjude.org

Neuroblastoma is the most common extracranial solid tumor of childhood[1]. *MYCN* amplification and age at diagnosis are the two most powerful predictors of outcome, with survival rates 5–10 times higher in infants than in adolescents or young adults[1,2]. Previous genomic analyses of stage 4 pediatric neuroblastoma samples identified the *ATRX* mutations in patients that were typically older than 5 y, had an indolent disease course, and poor overall survival (OS)[1,3].

One important function of ATRX is recognition of guanine (G)-rich stretches of DNA and deposition of the H3.3 histone variant to prevent the formation of G-quadruplex (G4) structures, which can block DNA replication or transcription[4,5]. G-rich repeats are also found at telomeres and centromeres; ATRX forms a complex with DAXX to deposit H3.3 in those regions to maintain their integrity[4,5]. In cells lacking ATRX, H3.3 is not efficiently deposited at the telomeric G-rich regions, G4 structures form, and replication forks stall[4,5]. Consequently, telomeres undergo homologous recombination leading to alternative lengthening of telomeres (ALT)[6].

The formation of G4 structures in other G-rich repetitive regions of the genome can cause replicative stress[7,8] or block transcription[9]. Indeed, H3.3 is deposited at actively transcribed genes in addition to telomeres and pericentromeric DNA[9]. ATRX may also affect transcription by targeting the PRC2 complex to particular regions of the genome[10]. Consequently, in ATRX-deficient cells, PRC2-mediated modification of H3 to H3K27me3 lacks specificity, and genes that are normally repressed by polycomb are deregulated[10].

MYCN regulates diverse cellular processes during development and in cancer. For example, elevated MYCN leads to increased glycolytic flux and glutaminolysis to promote metabolic reprogramming associated with tumorigenesis in a variety of cancers including neuroblastomas[11,12]. MYCN-induced glutaminolysis in neuroblastoma elevates reactive-oxygen species (ROS) and DNA-replicative stress[13,14]. Indeed, one of the hallmarks of neuroblastoma is the DNA mutation signature associated with ROS induced DNA damage. Consequently, neuroblastoma cells exhibit increased sensitivity to pharmacological agents that induce oxidative stress[13,14]. Here we demonstrate that the DNA-replicative stress induced by *ATRX* mutations and *MYCN* amplification cause synthetic lethality in neuroblastoma. This is unusual because oncogene activation and tumor-suppressor inactivation often work in concert to promote tumorigenesis not cancer cell death.

## Results

### *ATRX* and *MYCN* mutations in neuroblastoma.

To complement previous neuroblastoma studies from the Therapeutically Applicable Research to Generate Effective Treatment (TARGET) initiative[15] and the Pediatric Cancer Genome Project (PCGP)[3,16], we obtained neuroblastoma samples from 473 patients (122 unpaired and 351 paired tumor/germline) from the Children's Oncology Group (COG) (Table 1). We identified single-nucleotide variations, small indels, and other somatic mutations in the coding region of *ATRX* via custom capture and Illumina sequencing using probes spanning the entire *ATRX* locus and whole-exome sequencing of 828 germline and tumor samples. We also included *MYCN* in the capture probe set to determine its copy number. We identified 19 somatic *ATRX* mutations, 15 of those are internal deletions predicted to encode truncated proteins (Fig. 1a).

Significantly higher *ATRX*-mutation frequencies are detected in patients with International Neuroblastoma Staging System (INSS) stage 4 disease (8.6%), high-risk subgroup (14.6%), 11q loss of heterozygosity (LOH), or unfavorable histology and in

**Table 1 Distribution of *ATRX* mutations in neuroblastoma.**

| Classification[a] | Grouping | N(%)[b] | *ATRX* mutation (%)[c] | *p*-value |
|---|---|---|---|---|
| INSS stage | Non-stage 4 | 312 (65.8) | 5 (1.6) | 0.0002 |
| | Stage 4 | 162 (34.2) | 14 (8.6) | |
| Risk Group | Low/intermediate | 375 (79.6) | 5 (1.3) | 4.4e-7 |
| | High | 96 (20.4) | 14 (14.6) | |
| Sex | Female | 253 (53.4) | 12 (4.7) | 0.3830 |
| | Male | 221 (46.6) | 7 (3.2) | |
| *MYCN* (FISH) | Not amplified | 431 (91.7) | 18 (4.2) | 1.0000 |
| | Amplified | 39 (8.3) | 1 (2.6) | |
| *MYCN* (NGS) | Not amplified | 447 (94.1) | 19 (4.3) | 0.6180 |
| | Amplified | 28 (5.9) | 0 (0) | |
| *ALK* | No mutation | 454 (95.6) | 17 (3.7) | 0.2026 |
| | Mutation | 21 (4.4) | 2 (9.5) | |
| Ploidy | Hyperdiploid | 277 (60.7) | 10 (3.6) | 0.6454 |
| | Diploid | 179 (39.3) | 8 (4.5) | |
| 11q LOH | No | 301 (91.8) | 5 (1.7) | 4.9e-4 |
| | Yes | 27 (8.2) | 5 (18.5) | |
| 1p LOH | No | 290 (88.4) | 9 (3.1) | 1.0000 |
| | Yes | 38 (11.6) | 1 (2.6) | |
| Histology | Favorable | 314 (69.8) | 1 (0.3) | 9.3e-9 |
| | Unfavorable | 136 (30.2) | 17 (12.5) | |
| Age at diagnosis | <18 months | 213 (44.9) | 0 (0) | 0.00001 |
| | ≥18 months | 261 (55.1) | 19 (7.3) | |
| Age at diagnosis | <18 months | 213 (44.9) | 0 (0) | 9.0e-6 |
| | 18 mo-5 yrs | 76 (16.0) | 1 (1.3) | |
| | 5–12 yrs | 152 (32.1) | 15 (9.9) | |
| | >12 years | 33 (7.0) | 3 (9.1) | |
| Grade | Differentiating | 43 (13.2) | 3 (7.0) | 0.7168 |
| | Poorly/undifferentiated | 283 (86.8) | 15 (5.3) | |
| MKI | Low/intermediate | 298 (90.3) | 18 (6.0) | 0.2356 |
| | High | 32 (9.7) | 0 (0) | |
| Race | Black | 55 (12.7) | 5 (9.1) | 0.0348 |
| | other | 378 (87.3) | 12 (3.2) | |

[a]INSS international neuroblastoma staging system, *MYCN* amplification was assessed by FISH at each COG participating center and by next-generation sequencing (NGS) in our sequence analysis, *MKI* mitosis–karyorrhexis index.
[b]N number of patients with percentage in parentheses.
[c]There were 19 *ATRX* mutations identified but for some tumors, the classifier was not applied so the total number of samples and the total number of ATRX-mutant samples is lower.

those who are 18 mo or older at diagnosis (Table 1, Fig. 1b). *ATRX*-mutant neuroblastomas were significantly more likely to have ALT than *ATRX* wild-type tumors (89.5% (17/19) vs. 22.2% (4/18); $p < 0.0001$, two-tailed Fisher's exact test). In addition, black patients had a significantly higher frequency of *ATRX* mutation than did those in other racial categories (9.1% vs. 3.3%; $p = 0.0348$, two-tailed Fisher's exact test) (Table 1). *ATRX* mutation frequency did not differ based on sex, *ALK* status, tumor grade, mitosis–karyorrhexis index, or ploidy (Table 1).

*ATRX* mutations were associated with significantly lower 4-year event-free survival (EFS) rates among the groups with non–INSS stage 4 disease ($p = 0.0007$, two-tailed Fisher's exact test), INSS stage 4 disease ($p = 0.0128$, two-tailed Fisher's exact test), ages 5–12 y ($p = 0.0006$, two-tailed Fisher's exact test) and older than 12 y ($p = 0.0038$, two-tailed Fisher's exact test) at diagnosis in the COG cohort (Supplementary Data 1 and Fig. 1c, d). Compared with those without a mutation, patients harboring an *ATRX* mutation were 5.0 times more likely to experience an adverse event ($p < 0.0001$) and 3.4 times more likely to die of their disease ($p = 0.0046$, two-tailed Fisher's exact test) (Supplementary Data 1).

In the cohort of 819 tumors, *ATRX* mutations ($n = 64$) and *MYCN* amplification ($n = 140$) showed statistically significant mutual exclusivity ($p = 0.0037$, Cochrane-Mantel-Haenszel test) (Fig. 1e and Supplementary Note 1). Only 1 of 819 (0.1%) tumors had both lesions. However, this was an unusual discordant sample because it was scored as *MYCN*-amplified by fluorescence in situ hybridization (FISH) in COG's neuroblastoma reference lab but non-amplified based on normalized read depth from our custom capture Illumina next-generation sequencing data (Supplementary Fig. 1A). To resolve this discrepancy, we repeated

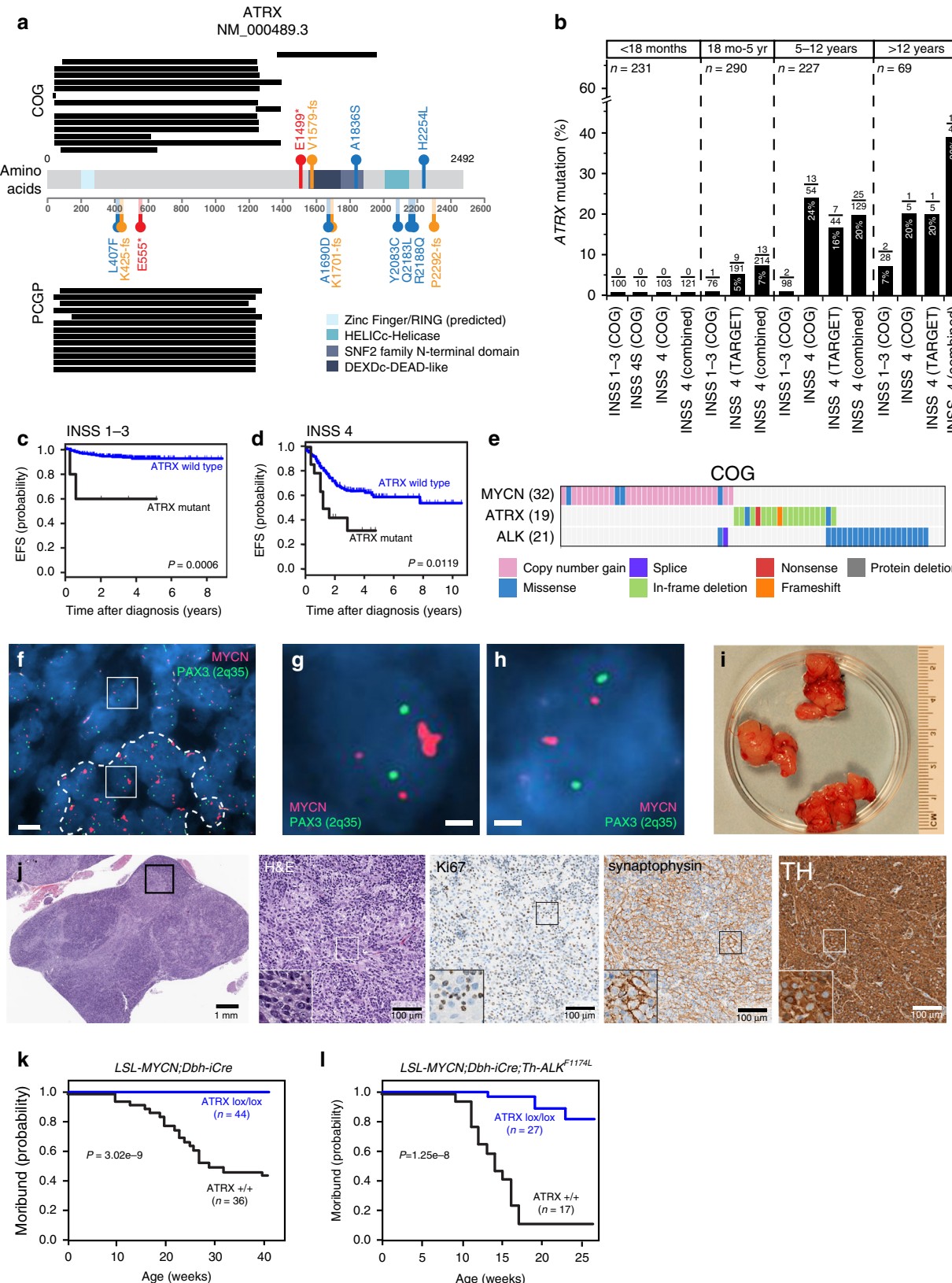

the FISH analysis and found regional heterogeneity of *MYCN* amplification (Fig. 1f–h). Specifically, the region with *MYCN* amplification had a high mitosis–karyorrhexis index and was less likely to have ultrabright telomere foci, characteristic of ALT, in

*ATRX*-mutant neuroblastoma cells (Fig. 1f–h and Supplementary Fig. 1B–E).

To determine whether *ATRX* mutations and *MYCN* amplification are incompatible in vivo, we conditionally inactivated *ATRX*

**Fig. 1 ATRX mutations are mutually exclusive of MYCN amplification in neuroblastoma. a** Summary of *ATRX* mutations in the COG and PCGP cohorts. Black bars indicate the deleted amino acids; red indicates nonsense mutations; orange indicates frameshift mutations; and blue indicates missense mutations. The major protein domains are shaded in blue. **b** Bar plot of the percentage of the 819 patients analyzed in this study with *ATRX* mutations for each age and stage. **c, d** EFS for patients with neuroblastoma with or without *ATRX* mutations for stages 1–3 or stage 4 for all age groups. *P* values were calculated using Cox model. **e** Heatmap of the distribution of mutations in *MYCN*, *ATRX*, and *ALK* in the COG cohort. Only those tumors with a mutation in at least one of the three genes are indicated. **f** Micrograph of *MYCN* FISH (red) for the one neuroblastoma sample with an *ATRX* mutation and *MYCN* amplification showing regional amplification (dashed line). The PAX3 (2q35) probe (green) is the control. **g** Micrograph of a neuroblastoma cell with *MYCN* homogenously staining region within a region of the tumor that has *MYCN* amplification (box within the dashed line region in **f**. **h** Micrograph of a neuroblastoma cell with two copies of *MYCN* outside of the region with amplification (box outside of the dashed line region). **i** Representative photograph of neuroblastoma tumors from a *LSL-MYCN;Dbh-iCre* mouse. All the tumors developed in this mouse cohorts were harvested and examined. **j** Micrographs of the histology of the tumor shown in **i**. The boxed region in each micrograph is magnified in the lower left panel of each image. Three tumors from each group of mice were examined by both H&E and immune-staining. **k, l** Survival curves for two mouse models of neuroblastoma with or without conditional *ATRX* deletion. The numbers of mice in each group are indicated. *P* values were calculated using Log-rank test. Scale bar: **f**, 5 µm, **g**, **h**, 1 µm. *COG* Children's Oncology Group, *PCGP* Pediatric Cancer Genome Project, *INSS* International Neuroblastoma Staging System, *H&E* hematoxylin and eosin, *TH* tyrosine hydroxylase.

in two genetically engineered neuroblastoma mouse models (Fig. 1i–l)[8,17,18]. One model (*LSL-MYCN;Dbh-iCre*) is a transgenic line with conditional expression of *MYCN* from *Rosa26a* locus by Cre expressed in the dopaminergic cells of the sympathoadrenal lineage[19]. The other model includes the *Th-ALK^{F1174L}* transgene that potentiates *MYCN*-mediated tumorigenesis[20]. For each of these models, tumor formation was reduced, and survival was significantly increased when *Atrx* was simultaneously inactivated with elevated *MYCN* expression (Fig. 1k–l). The few tumors that did form in the *Atrx^{Lox/Lox};LSL-MYCN;Dbh-iCre;ALK^{F1174L}* mice all had evidence of an intact *Atrx* allele, suggesting that rare cells that did not undergo *Atrx* inactivation contributed to tumor formation. It is also possible that ALK signaling partially rescued the synthetic lethality. *Atrx* inactivation has been shown to reduce cell survival in brain, muscles and testes[8,21,22]. Histopathological review of adrenal gland and paravertebral sympathetic ganglia from *LSL-MYCN;Dbh-iCre* and *Atrx^{Lox/Lox};LSL-MYCN;Dbh-iCre* mice at 3 weeks and 1 year of age showed no differences (Supplementary Fig. 2, Supplementary Note 2), suggesting that the lack of tumor formation was not due to death of the NB cell of origin.

**ATRX mutations are incompatible with elevated MYCN.** To extend our data to humans, we characterized the expression of MYCN, ATRX, and DAXX proteins across 12 human cancer cell lines (Fig. 2a). We used eight neuroblastoma cell lines, including three *MYCN*-amplified lines (IMR32, SKNBE2, NB-5) or moderately elevated levels of MYCN (NBL-S) (Supplementary Data 2), two lines (SKNMM and CHLA90) produce ATRX with an in-frame deletion (ATRX^{IFD}) and are diploid for *MYCN* (Fig. 2a), one osteosarcoma cell line with no detectable expression of ATRX (U2OS) and three lines as controls.

To knockdown ATRX, we used two different lentiviral vectors expressing shRNAs targeting *ATRX* (#9 and #20) (Fig. 2b). The optimal knockdown was ~ 86% in HeLa cells after 48 h with *ATRX* shRNA–20. Two weeks after introducing shRNAs, telomere content, ALT status and the proportions of cells in different phases of the cell cycle did not differ in any cell line tested (Supplementary Fig. 3A–I). Knocking down ATRX decreased colony formation in *MYCN*-amplified neuroblastoma cells (IMR32 and NB-5) but not cell lines expressing wild-type *MYCN* (Fig. 2c, d).

To inactivate ATRX in the *MYCN*-amplified neuroblastoma cell lines, we designed and validated two guide RNAs (gRNA-2 and gRNA-5) to exon 6 of *ATRX* (Fig. 2e, f). We transfected 293T, SKNBE2, and IMR32 cells with each gRNA and a plasmid expressing Cas-9 and purified the transfected cells by flow cytometry. At 2, 10, and 14 days after plating the purified cells, we harvested DNA, performed PCR using primers spanning both of the two gRNA-target sequences, and performed Illumina MiSeq analysis to calculate the proportion of mutated alleles in each

sample. Although the *ATRX*-mutant alleles were abundant in 293T and SKNBE2 cells 2 days after plating, those cells harboring *ATRX*-mutant alleles were lost by 10 days in culture (Fig. 2g and Supplementary Fig. 3J). In *MYCN*-amplified IMR32 cells, no mutant alleles were detected at any timepoint, though a control gRNA to the *AAVS1* locus led to efficient mutagenesis. A similar study performed with a gRNA to *DAXX* had no effect on the viability of 293T, SKNBE2, or IMR32 cells (Supplementary Fig. 3K-M). Taken together, our data suggest that *ATRX* inactivation is incompatible with *MYCN* amplification in neuroblastoma, which is consistent with data from the Project Achilles (https://depmap.org/portal/achilles/).

Next, we induced the expression of *MYCN* in *ATRX*-mutant neuroblastoma cells by using a doxycycline-inducible vector (Fig. 2h). Initially, we used two *ATRX*-mutant cell lines (SKNMM and U2OS), a *MYCN*-amplified cell line (SKNBE2), and a cell line that is wild-type for both genes (SKNFI) (Fig. 2a). The levels of MYCN achieved with doxycycline induction in both *ATRX*-mutant lines were similar to those in *MYCN*-amplified cell lines (Fig. 2h).

The growth rates of SKNFI^{MYCN} and SKNBE2^{MYCN} cells, after 10 days in culture ± doxycycline, showed no difference with induction of MYCN expression, but by 6 days in culture, the two *ATRX*-mutant lines showed a marked loss of cells (Fig. 2i, j). A parallel experiment performed with stable cell lines that had only the doxycycline-inducible vector lacking *MYCN* showed no effect on cell growth. More than 95% of the cells in the SKNMM^{MYCN} cultures were lost by 8–10 days in the presence of doxycycline (Fig. 2k). Live imaging of cells ± doxycycline on Days 6–8, the peak period of cell loss (Fig. 2l), enabled us to determine the timing of cell death in 44 cells from 13 different movies (Supplementary Data 3). In total, 54% of the cells died after initiating cytokinesis; the remaining cells died $20.2 \pm 12.8$ h (mean ± standard deviation) after cytokinesis. These data were extended to five other *ATRX*-mutant cell lines including two glioma cell lines that had isogenic *ATRX*-wild-type controls (Supplementary Fig. 4)[23]. A marker of DNA-replicative stress (pRPA32) was upregulated in the absence of ATRX and in the presence of doxycycline. There was no effect of ectopic MYCN expression on a DAXX-mutant cell line (Supplementary Fig. 4A, B). A small number of SKNMM^{MYCN} cells escaped cell death and continued to grow for 45 days in the presence of doxycycline. The ATRX^{IFD} was still expressed and the *MYCN* transgene was still present, but MYCN protein was lost (Fig. 2m).

To determine whether an elevated level of MYCN is incompatible with *ATRX* mutation in human neuroblastoma cells in vivo, we performed an in vivo growth competition assay. In one cohort of mice, SKNMM^{MYCN} cells labeled with luciferase and YFP were mixed in a 1:1 ratio with unlabeled control SKNMM^{CONT} cells,

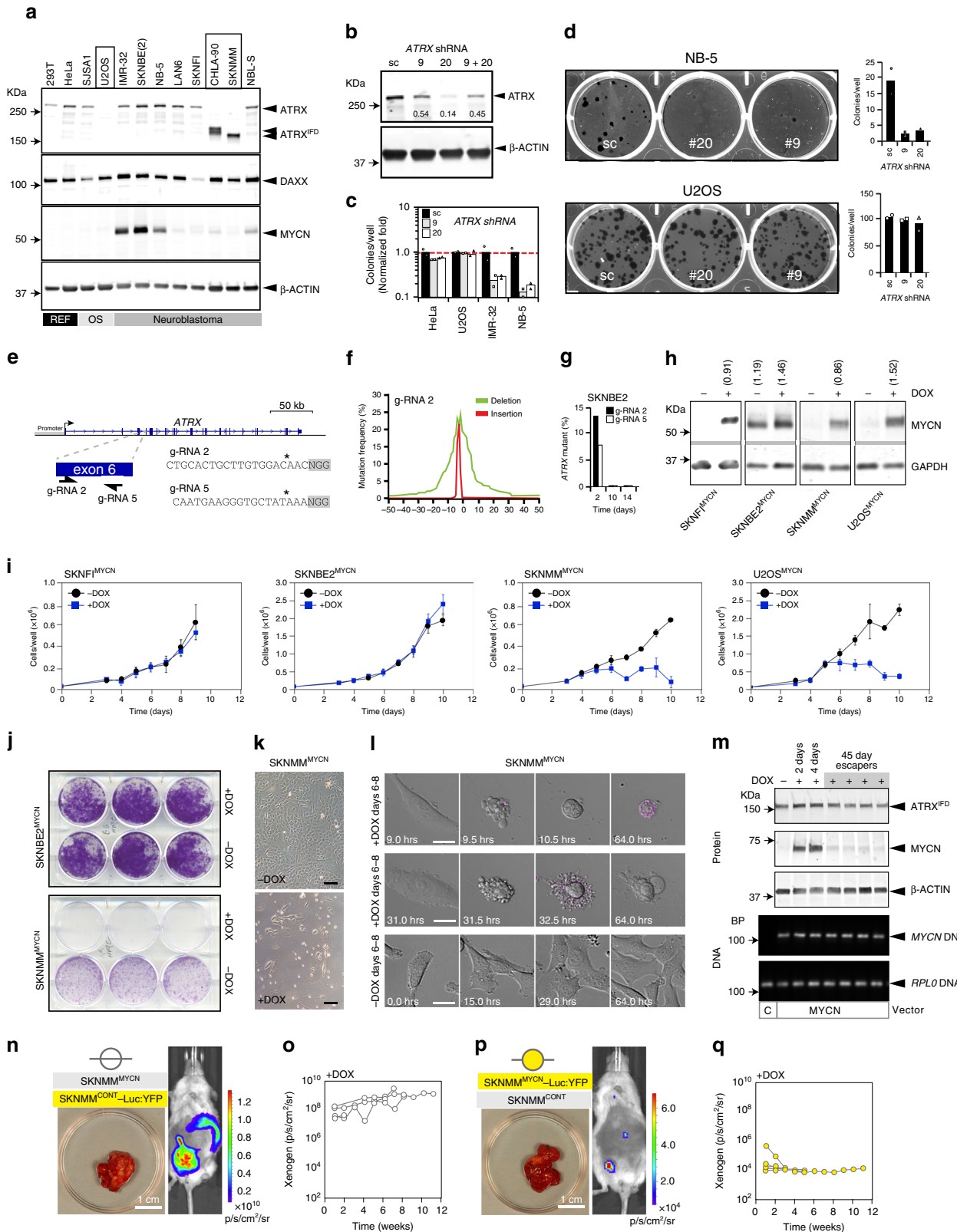

which contained the doxycycline-inducible vector lacking *MYCN*. In a second cohort, the SKNMM[CONT] cells were labeled with luciferase and YFP and the SKNMM[MYCN] cells were unlabeled. Cells were injected into the para-adrenal space of 25 immunocompromised mice for each cohort. The luciferase was monitored every

week, and ultrasound was performed every 2 weeks. Animals were randomized to +doxycycline and −doxycycline groups once the tumor was larger than 14 mm³ by three-dimensional ultrasound and/or $1.4 \times 10^7$ photons/s/cm²/sr by xenogen imaging (Supplementary Data 4). The SKNMM[MYCN] cells were outcompeted by the

**Fig. 2 *ATRX* mutations and *MYCN* amplification are incompatible. a** Immunoblot of 12 cell lines. Boxes indicate *ATRX*-mutant lines. **b** Immunoblot for HeLa cells transfected with sc-shRNA or *ATRX*-specific shRNAs. The fraction of ATRX relative to sc-shRNA is indicated. **c** Bar plot of the number of colonies/well (mean of duplicates) after transfection of shRNAs, normalized to the sc-shRNA (dashed line). **d** Photograph of cresyl violet-stained colonies after transfection with shRNAs and bar plots of the average number of colonies/well. **e** Map of two gRNAs targeting *ATRX*. **f** Plot of the mutation frequency for gRNA-2, deletions (green) and insertions (red). **g** Bar plot of the proportion of *ATRX*-mutant alleles (100,000× coverage/sample). **h** Immunoblot for the cell lines with the doxycycline-inducible *MYCN* expression construct. Numbers indicate MYCN/GAPDH fluorescence intensity. The experiment was done three times with similar results. **i** Line graphs of the growth curves (mean ± SD) for each cell line ± doxycycline, $n = 3$. **j** Colony assay for SKNBE2$^{MYCN}$ and SKNMM$^{MYCN}$ cells ± doxycycline, the experiment was repeated twice with similar results. **k** Brightfield micrograph of SKNMM$^{MYCN}$ cells after 8 days in culture ± doxycycline, the experiment was repeated twice with similar results. **l** Brightfield micrographs of three individual SKNMM$^{MYCN}$ cells ± doxycycline at indicated timepoints, relative to the starting point of the movies (6 days + doxycycline), 44 cells were analyzed. **m** Immunoblots of SKNMM$^{MYCN}$ cells without doxycycline or after 2, 4, or 45 days in culture. The escapers are pools of cells that grew in the presence of doxycycline ($n = 4$). PCR for *MYCN* transgene and a control locus (*RPL0*) is indicated in the lower portion. **n** Xenogen image of a mouse with an orthotopic neuroblastoma tumor (photograph), that arose from a 1:1 mixture of SKNMM$^{MYCN}$ cells and SKNMM$^{CONT}$–Luc:YFP cells ($n = 5$). **o** Line graph of xenogen image data described in **n**. **p** Xenogen image of a mouse with an orthotopic neuroblastoma tumor (photograph), that arose from a 1:1 mixture of SKNMM$^{MYCN}$–Luc:YFP cells and SKNMM$^{CONT}$ cells ($n = 5$). **q** Line graph of xenogen image data described in **p**. Scale bars: **k**, 10-μm. *DOX* doxycycline, *IFD* in-frame deletion, *sc* scrambled.

SKNMM$^{CONT}$ cells in vivo, as determined by xenogen and ultrasound (Fig. 2n–q).

**The epigenomic landscape of *MYCN*-amplified and *ATRX*-mutant NB**. MYCN is a global transcriptional regulator that activates and represses genes controlling cell division, cell size, and cell differentiation during development[19]. MYCN can change gene expression by modifying the epigenetic status of target genes[19]. To analyze the epigenomic landscape in *ATRX*-mutant neuroblastoma cells, compared with that in wild-type *ATRX* neuroblastoma, we selected eight cell lines that encompass the major genetic groups, including those with *MYCN* amplification or *ATRX* mutations (Fig. 3a). We also included eight orthotopic patient-derived xenografts (O-PDXs) that were derived by para-adrenal injection of patient tumors into immunocompromised mice as described previously[20,24] (Fig. 3a). One of those O-PDXs (SJNBL047443_X1) has an *ATRX* in-frame deletion (Fig. 3b, c and Supplementary Data 2). Autopsy tumor material from a patient with *ATRX*-mutant neuroblastoma (SJNBL030014_D) was also included (Fig. 3a and Supplementary Data 2). We performed whole-genome bisulfite sequencing (WGBS; Supplementary Fig. 5 and Data 5), RNA-sequencing, and ChIP-seq of eight histone marks (H3K4me1, H3k4me2, H3K4me3, H3K27me3, H3K27Ac, H3K36me3, H3K9/14Ac, and H3K9me3), CTCF, BRD4, and RNA polymerase II (PolII) across all eight cell lines, the eight O-PDX tumors, the autopsy sample, and normal fetal adrenal medulla. We also performed MYCN ChIP-seq on three O-PDXs with *MYCN* amplification (SJNBL046_X, SJNBL012407_X1, and SJNBL013762_X1) and the SKNFI$^{MYCN}$, SKNBE2$^{MYCN}$, and SKNMM$^{MYCN}$ cell lines with doxycycline on Day 4 in culture as well as matched controls. To relate gene expression changes to epigenetic changes induced by MYCN, we performed ChIP-seq for the eight histone marks, CTCF, Brd4, and RNA PolII in the SKNMM$^{MYCN}$ cells ± doxycycline on Day 4 in culture.

To define the chromatin states and analyze the transitions thereof across the genome, we performed chromatin Hidden Markov Modeling (ChromHMM) using all 393 ChIP-seq data sets (Fig. 3d, e and Supplementary Data 6). To share these data, we have developed a cloud-based viewer (https://pecan.stjude.org/proteinpaint/study/mycn_nbl_2018). We identified 394 genes that were upregulated and 544 that were downregulated 4 days after adding doxycycline to the SKNMM$^{MYCN}$ or U2OS$^{MYCN}$ cell lines (Supplementary Data 7). Gene set-enrichment analysis of the upregulated genes identified MYC and MYCN targets as the most-enriched data sets (Supplementary Fig. 6A): 84% of the upregulated genes and 42% of the downregulated genes had a MYCN-binding site based on

MYCN ChIP-seq for SKNMM$^{MYCN}$ cells in the presence of doxycycline (Supplementary Fig. 6B-E and Data 7). Analysis of the chromHMM for the upregulated MYCN targets showed that expansion of state 13 (strong transcribed gene body) downstream of the transcriptional start site was the most significant change (Fig. 3f, g and Supplementary Data 7). This is consistent with previous data showing that MYC regulates RNA PolII pause release not transcriptional initiation[25]. These upregulated target genes are enriched for the pathways involved in metabolism (e.g., *SLC3A2, SLC7A5*) and mitochondrial gene expression (e.g., *PNPT1*) (Fig. 3f, g, Supplementary Fig. 6B-E, and Data 7), consistent with previous studies, demonstrating that elevated MYC/MYCN induces metabolic reprogramming in cancer cells[11].

**Metabolic reprogramming induced by MYCN**. One form of metabolic reprogramming induced by MYC/MYCN is glutamine addiction[26]. The glutamine transporter (SLC1A5) involved in glutamine uptake and the bidirectional transporter (SLC7A5–SLC3A2) involved in glutamine efflux coupled with essential amino acid import are direct targets of MYCN and are upregulated in SKNMM$^{MYCN}$ cells in the presence of doxycycline (Fig. 3g, Supplementary Data 7 and 6D, E)[27]. Following *MYCN* induction, we quantified glutamine metabolism in SKNMM$^{MYCN}$ cells ± doxycycline on Day 4 in culture and measured eight tricarboxylic acid (TCA) cycle cellular metabolites[28]. The proportions of five carbon-labeled glutamate and α-ketoglutarate increased, consistent with glutamine utilization for the TCA cycle as well as reductive carboxylation to produce five carbon-labeled citrate (Fig. 4a, b).

Inefficient use of glutamine and/or glucose by cancer cells can lead to the export of glycolytic intermediates such as lactate[29]. The pH of the medium was quickly acidified in subconfluent cultures of SKNMM$^{MYCN}$ or U2OS$^{MYCN}$ cells in the presence of doxycycline; however, SKNBE2$^{MYCN}$ and SKNFI$^{MYCN}$ cells showed no difference in the pH of their media (Fig. 4c, d). The reduced pH correlated with an increase in lactate in the culture medium and dependence on glutamine for survival (Fig. 4e and Supplementary Fig. 7A). To provide additional data on the metabolic reprogramming induced by MYCN in *ATRX*-deficient cells, we performed a more-comprehensive metabolomic profiling of 54 metabolites using $^{13}$C-labeled glucose and $^{13}$C-labeled glutamine (Fig. 4f, g and Supplementary Data 8). Glucose was the major source of lactate; relatively few of the carbons from glucose were used for the TCA cycle (Fig. 4f, g and Supplementary Data 8). In contrast, glutamine was converted to α-ketoglutarate for the TCA cycle and reductive carboxylation.

Glutamine is also an important mitochondrial substrate; cells must precisely balance the expression and activity of proteins

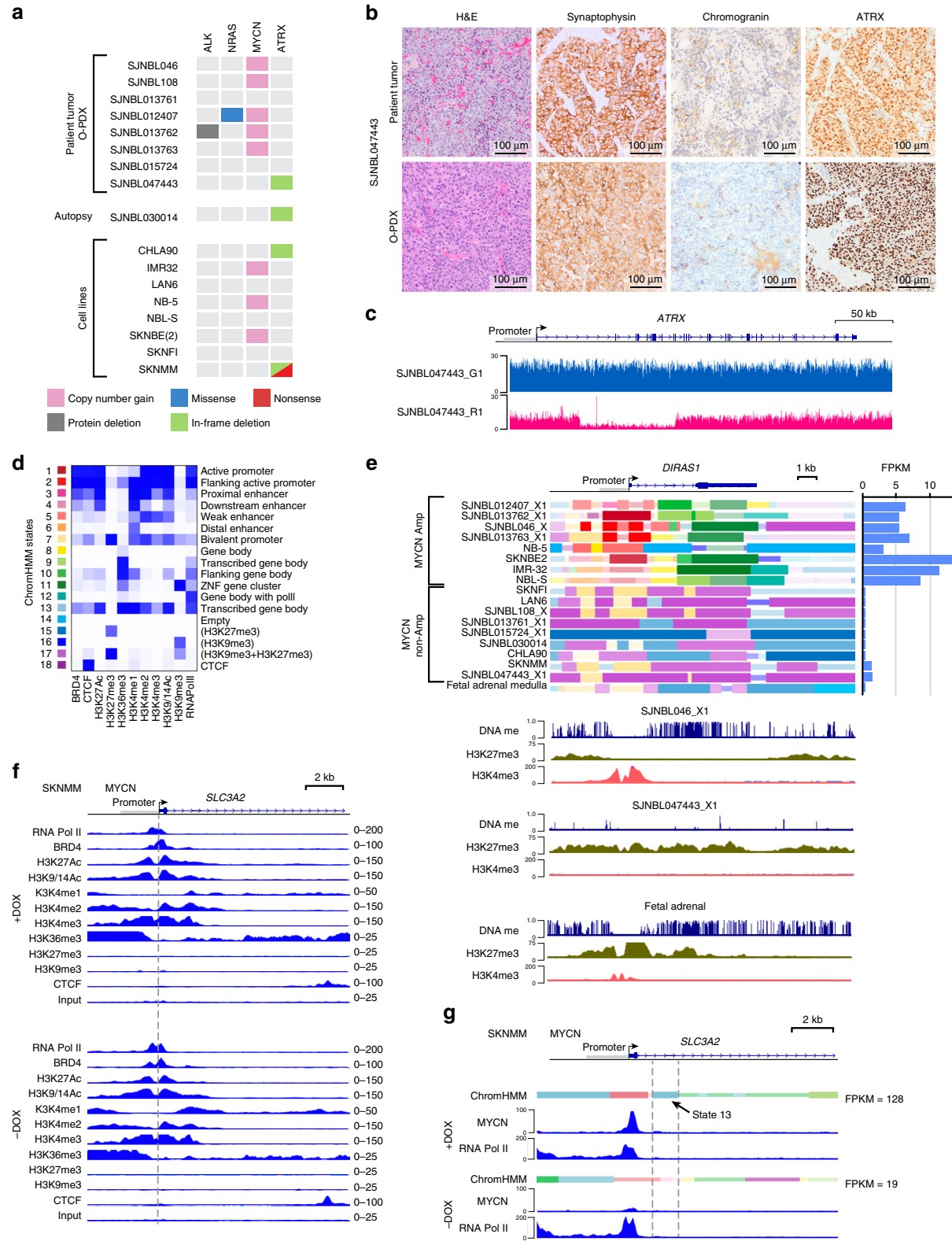

encoded by the nucleus with those encoded by the mitochondria to maintain homeostasis[28]. Cells also upregulate pathways required to mitigate the ROS that are a natural byproduct of mitochondrial metabolism to prevent excessive protein or DNA damage. For example, glutamine uptake in cancer cells can elevate glutathione levels because glutamine is converted to the glutathione precursor, glutamate[30]. Cell lines that were sensitive to glutamine depletion in the medium also had significantly

elevated levels of glutathione (Supplementary Fig. 7B). ROS and DNA damage were increased in the absence of glutamine (Supplementary Fig. C, D).

Pathways involved in mitochondrial homeostasis were upregulated in SKNMM^MYCN or U2OS^MYCN cells in the presence of doxycycline (Supplementary Data 7). Little change in mitochondrial mass was measured using MitoTracker Green in SKNMM^MYCN or U2OS^MYCN cells ± doxycycline, but the mitochondrial membrane potential (Δψ_m)

**Fig. 3 Transcriptional and epigenetic changes induced by MYCN in *ATRX*-deficient neuroblastoma cells. a** Heatmap of the mutations in the O-PDXs and cell lines used in this study. An autopsy sample was also used for molecular profiling. **b** Micrographs of histologic analysis of the *ATRX*-mutant neuroblastoma O-PDX and corresponding patient tumor. **c** Plot of sequence read depth from Illumina sequencing for the *ATRX* gene in the patient's germline (blue) and tumor (red), showing somatic deletion of exons 2–11. **d** Heatmap of the chromHMM states used in this study with functional annotation. **e** ChromHMM plot of the *DIRAS1* gene and a bar plot of the gene's expression on the right. DNA methylation and H3K27me3 and H3K4me3 ChIP-seq for *DIRAS1* for a *MYCN*-amplified tumor (SJNBL046_X1), an *ATRX*-mutant tumor (SJNBL047443_X1), and fetal adrenal medulla. **f** ChIP-seq tracks for a portion of the *SLC3A2* gene spanning the promoter in SKNMM$^{MYCN}$ cells in the presence or absence of doxycycline after 4 days in culture. Dashed line indicates the start of transcription. The scales are indicated on the right. **g** ChromHMM and ChIP-seq for MYCN and RNA PolII for a portion of the *SLC3A2* gene spanning the promoter in SKNMM$^{MYCN}$ cells in the presence or absence of doxycycline after 4 days. The dashed lines indicate the region with induction of ChromHMM state 13 that is expanded after MYCN binds the promoter. *ChromHMM* chromatin Hidden Markov Modeling, *FPKM* fragments per kilobase of transcript per million mapped reads, *O-PDX* orthotopic patient-derived xenograft.

measured with tetramethylrhodamine-ethyl ester (TMRE) was significantly lower in the presence of doxycycline (Fig. 4h, i). To assess mitochondrial function in MYCN-overexpressing SKNMM cells, we measured the oxygen consumption rate. Induction of MYCN in SKNMM cells decreased basal mitochondrial respiration and respiratory reserve capacity, relative to that seen in un-induced SKNMM cells (Fig. 4j, k and Supplementary Fig. 8). Transmission electron micrographs showed that SKNMM$^{MYCN}$ cells in the presence of doxycycline had more disrupted mitochondria on Day 4 and subsequent timepoints than did SKNMM$^{MYCN}$ cells maintained in culture without doxycycline or the other cell lines (Fig. 4l–o, Supplementary Fig. 9). Similar results were obtained for the *ATRX*-deficient U2OS$^{MYCN}$ cells but not *ATRX*-wild-type SKNBE2-$^{MYCN}$ cells (Supplementary Fig. 9).

**Replicative stress in *ATRX*-mutant neuroblastomas expressing MYCN.** The mitochondrial dysfunction described above may lead to the accumulation of ROS and in turn contribute to replicative stress through DNA damage. The ROS increase began at Day 4 and persisted until Day 6 in SKNMM$^{MYCN}$ cells in the presence of doxycycline (Fig. 5a). This was accompanied by increased expression of γH2AX protein, a biomarker for DNA double-strand breaks (Fig. 5b, c). Spectral karyotyping showed a significant increase in cells with DNA fragmentation and translocation in the SKNMM$^{MYCN}$ cells in the presence of doxycycline (Fig. 5d). We also detected increased DNA fragmentation in SKNMM$^{MYCN}$ and U2OS$^{MYCN}$ cells in the presence of doxycycline by using a single-cell gel electrophoresis assay (Fig. 5e–g).

Cells with DNA-replicative stress are sensitive to hydroxyurea[31]. Thus, we maintained SKNMM$^{MYCN}$ cells cultured ± doxycycline with different concentrations of hydroxyurea (12.5, 25, 50, or 100 μM on Days 1–4) and measured cell viability using CellTiter-Glo on Day 4. The SKNMM cells expressing MYCN were significantly more sensitive to hydroxyurea at 50 or 100 μM concentrations (Supplementary Fig. 10A). The DNA content in each cell line, ±doxycycline at Day 4 in culture, showed an increase in S and G2/M phases in the SKNMM$^{MYCN}$ cells in the presence of doxycycline, which was consistent with cell cycle arrest as a result of DNA damage (Supplementary Fig. 10B). The ROS scavenger *N*-acetyl cysteine had a small but reproducible effect on reducing cell death induced by ectopic MYCN expression in SKNMM cells (Supplementary Fig. 10C, D).

To identify other pathways that may modulate to MYCN-induced cell death in *ATRX*-deficient cells, we performed a dose–response screen using two drug libraries. The first was a collection of 177 oncology drugs and compounds in late clinical development (Phase II or later). The second was a customized library of 34 drugs and compounds that target proteins or pathways that have been reported to reduce MYC/MYCN oncogenic activity (Data S9). Within the oncology drug set, 37 compounds selectively potentiated killing of SKNMM$^{MYCN}$ cells

in the presence of doxycycline (Supplementary Data 9); the majority (59.4%, 22/37) were drugs that induce DNA damage, inhibit DNA repair or replication, or inhibit cell cycle checkpoints (Supplementary Fig. 10E–G and Data 9). The only two drugs that partially rescued the cell death were a third-generation retinoid derivative (bexarotene) and an mTORC1/2 inhibitor (INK128) (Supplementary Fig. 10H and Data 9). The bexarotene result was particularly interesting because retinoids are an effective treatment for neuroblastoma (i.e., they reduce MYCN expression and induce differentiation). These results were independently validated using retinoic acid (Fig. 5h, i).

Together, these data suggest that MYCN induction causes metabolic reprogramming and mitochondrial dysfunction that contribute to increased ROS and DNA-replicative stress. It may be synthetically lethal in *ATRX*-mutant cells because they already have DNA-replicative stress in part owing to reduced ability to resolve G4 DNA structures. A subset of genes that are normally induced in *MYCN*-amplified neuroblastomas also may have failed to be activated in the *ATRX*-mutant neuroblastomas owing to the DNA and/or histone modifications. We identified one gene that met those criteria. *CUX2* encodes a homeodomain protein with three CUT repeats that is expressed in the developing nervous system and important for the repair of oxidative DNA damage[32]. *CUX2* is a direct MYCN target and is upregulated in *MYCN*-amplified neuroblastomas but not *ATRX*-mutant neuroblastomas (Fig. 5j). The *CUX2* promoter is epigenetically repressed by H3K27me3 in *ATRX*-mutant neuroblastomas and not induced in SKNMM cells in the presence of doxycycline (Fig. 5j). There was a partial rescue of cell death in SKNMM$^{MYCN}$ cells in the presence of doxycycline when CUX2 was ectopically expressed (Fig. 5k), suggesting that CUX2 is important for oxidative stress in *MYCN*-amplified neuroblastomas. *CUX2* is not induced by retinoic acid in 11 different NB cell lines (Supplementary Fig. 11 and Data 10) so the mechanism of partial rescue by retinoic acid is distinct from that of *CUX2*. Combining NAC with ectopic expression of *CUX2* did not significantly increase survival over NAC or *CUX2* alone (Supplementary Fig. 11B).

**Heterochromatic H3.3 deposition is altered in *ATRX*-mutant NB.** ATRX has an important role in H3.3 deposition in chromatin[33]. In addition to telomeres and pericentric heterochromatin, ATRX helps to resolve G-quadruplex (G4) structures in the DNA by deposition of H3.3 at those sites across the genome[5,33]. This is important because G4 structures can inhibit DNA replication and transcription[33,34]. G4 structures often overlap with noncanonical DNA:RNA hybrids called R-loops because both structures are favored in G-rich regions of the genome under negative torsional tension (e.g., active promoters)[35,36]. In cancer cells, R-loops can contribute to DNA breaks and genome instability[37,38]. To determine whether the pattern of H3.3 deposition is altered in *ATRX*-mutant NB, we performed CUT&RUN[39–41] on six NB cell lines divided into three groups: *MYCN*-amplified cells (SKNBE2 and

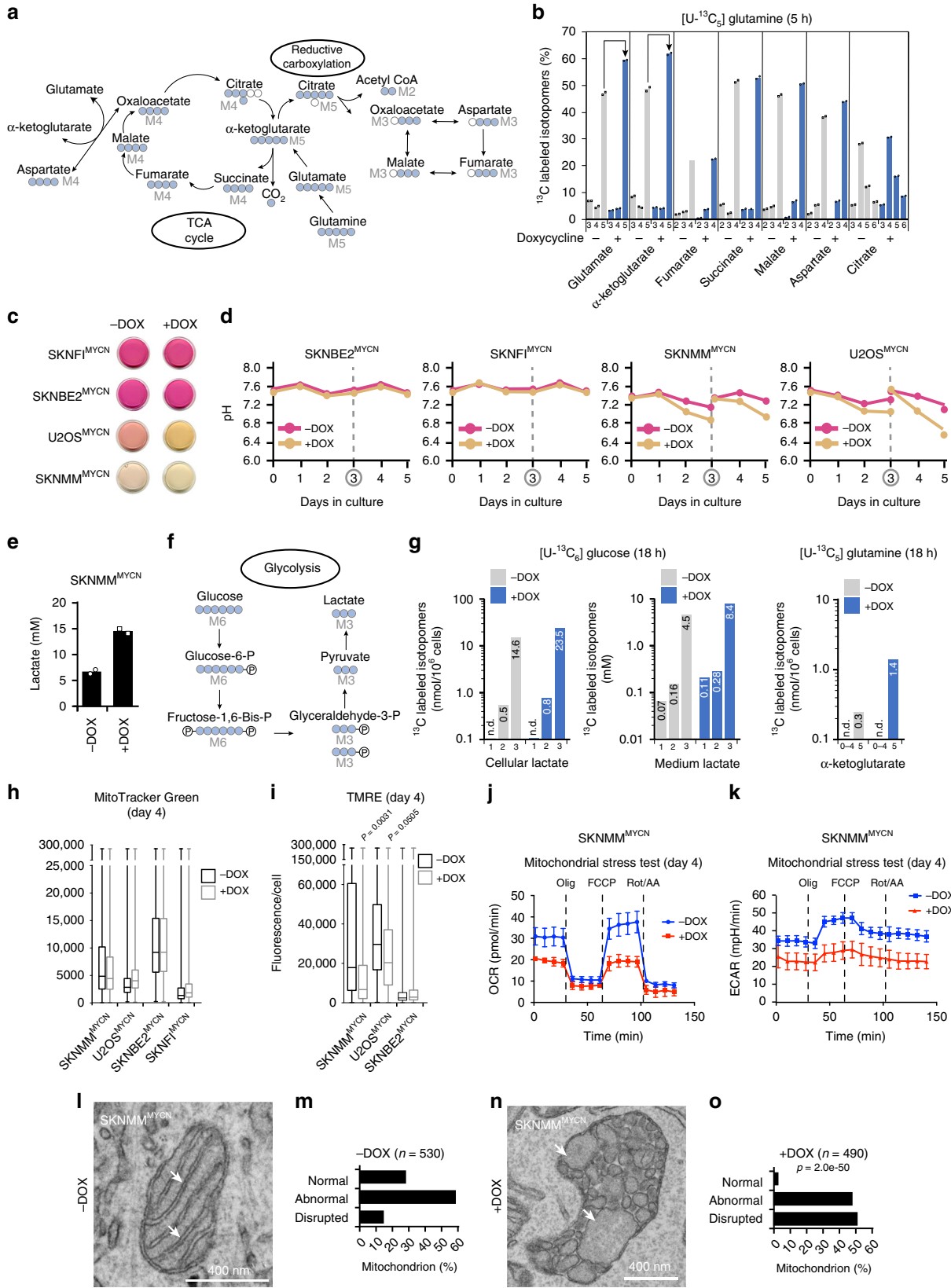

IMR32), *ATRX*-mutant cells (SKNMM and CHLA90) and *MYCN* non-amplified/*ATRX* wild-type cells (LAN6 and SKNFI). We identified the reproducible peaks in each group (ATRX, MYCN, and WT) and created a reference list of 51,919 H3.3 peaks across our data set. Among those peaks, 18,436 (35%) overlapped with

DNA sequences that are predicted to form G4 structures and 70% (12,932) of those are predicted to form R-loops[42]. As expected, there were fewer H3.3 peaks in the ATRX group and nearly a twofold reduction in H3.3 peaks overlapping G4 sequences (Fig. 6a). For every H3.3 peak in the data set, we assigned it to a

**Fig. 4 Expression of MYCN in *ATRX*-mutant cells leads to metabolic reprogramming and mitochondrial dysfunction. a** Simplified drawing of the TCA cycle and reductive carboxylation, blue circles: $^{13}C$ derived from $^{13}C_5$-glutamine. **b** Bar plot (mean of two technical replicates) of $^{13}C$-labeled isotopomers in SKNMM$^{MYCN}$ cells on day 4 ± doxycycline after 5 h of labeling with $^{13}C_5$-glutamine. The arrows indicate an increase in M + 5 glutamate and M + 5 α-ketoglutarate (the most abundant and direct derivatives of $^{13}C_5$-glutamine) in the presence of doxycycline. **c** Photograph of culture media on day 3 ± doxycycline. **d** Line plots of media pH ± doxycycline for each day. The culture media were changed on Day 3. **e** Histogram (mean of two biological replicates) of the levels of lactate at Day 4 in SKNMM$^{MYCN}$ cells ± doxycycline. **f** Simplified drawing of the glycolysis highlighting the production of lactate from $^{13}C_6$-glucose. **g** Bar plots of $^{13}C$-labeled isotopomers of lactate and α-ketoglutarate in the medium or cells after 18 h of labeling with $^{13}C_6$-glucose or $^{13}C_5$-glutamine, respectively, at Day 4 in SKNMM$^{MYCN}$ cells ± doxycycline, with concentrations indicated on the bars. **h, i** Box and Whiskers plots of MitoTracker Green (H) and TMRE (I) staining ± doxycycline on Day 4. Each Box shows the 10th to 90th percentiles range of data (at least 15,000 cells), the line represents the median and the Whiskers show the minimum and maximum data range. *P* values were calculated using two-tailed Mann–Whitney non-parametric test. **j, k** Representative results of oxygen consumption rate (OCR) and extracellular acidification rate (ECAR) (mean and SEM of three technical replicates) for $2.5 \times 10^4$ SKNMM$^{MYCN}$ cells grown under the Mito-Stress assay conditions for the Seahorse. The experiment was done three times with similar results. **l–o** Representative electron micrograph of mitochondria from SKNMM$^{MYCN}$ cells after 4 days ± doxycycline **l, n** showing disruption in the mitochondrial cristae (arrows). Bar plot of scoring of mitochondrial morphology for the SKNMM$^{MYCN}$ cells ± doxycycline, numbers of scored mitochondria are indicated **m, o**. *P* value was calculated using Chi-square test. *DOX* doxycycline, *FCCP* carbonyl cyanide-p-trifluoromethoxy phenylhydrazone, *Olig* oligomycin, *Rot/AA* rotenone/antimycin A, *TCA* tricarboxylic acid, *TMRE* tetramethylrhodamine ethyl ester.

group defined by the pattern across the samples (ATRX, MYCN, and WT). The constitutive (C) group includes H3.3 peaks found at the same location across all samples (ATRX, MYCN, and WT) (Fig. 6b). The enriched (E) group includes H3.3 peaks that are higher in ATRX than MYCN and WT (Fig. 6b). The depleted (D) group includes H3.3 peaks that are lower in ATRX than in MYCN and WT and the overlapping (O) group has peaks in ATRX and either MYCN or WT (Fig. 6b). The D group was the largest (Fig. 6b) and 2428 of those had a correlation between ChromHMM state and the presence or absence of H3.3 peaks; 514 (21%) had G4 sequences and 69% of those (357) were predicted to form R-loops. The majority of H3.3 peaks that were depleted in *ATRX*-mutant NB relative to MYCN and WT that had a correlation with ChromHMM state were in enhancers, whereas those with G4 sequences (and R-loops) were in promoters (Fig. 6c). For both the non-genic and genic regions, there was a shift from active euchromatic states (states 1–4) to inactive heterochromatic states (states 15–17) in the ATRX group relative to MYCN and WT (Fig. 6d–h). There was overlap between H3.3 peaks and CTCF sites (state 18) consistent with the role of H3.3 in CTCF binding to establish insulated chromatin domains[43–45]. We performed pathway analysis on the genes that had promoters or enhancers that were depleted for H3.3 in *ATRX*-mutant NB and had correlation with ChromHMM states and reduced expression in *ATRX*-mutant relative to MYCN and WT. The most significant pathways were involved in neuronal differentiation and neural development (Supplementary Data 11). For example, the *DUSP26* gene is a MYCN target that has G4 sequences immediately downstream of the promoter predicted to form R-loops and selective deposition of H3.3 in *MYCN*-amplified NB relative to *ATRX*-deficient NB cells (Fig. 6g). DUSP26 is a phosphatase that is expressed in neuroendocrine cells and can inhibit neuronal differentiation[46,47]. It has also been shown to dephosphorylate and inhibit p53 in neuroblastoma cells[48]. We performed the same H3.3 ChIP-seq on SKNMM$^{MYCN}$ with and without doxycycline and found similar chromatin changes at MYCN target genes (Fig. 6h). Taken together, our data show that there is marked reorganization of the chromatin landscape in *ATRX*-mutant NB beyond telomeres and centromeres. This is particularly notable at G4 sequences in promoters and enhancers and may impact the ability of *ATRX*-mutant NB cells to differentiate relative to those with wild-type ATRX.

## Discussion

Mutually exclusive mutation profiles are not uncommon in cancer cells; however, many are thought to result from targeting the same oncogenic pathway[49]. For example, inactivation of the *RB1* tumor-suppressor gene is often exclusive of amplification of

genes encoding cyclins (e.g., *CCNE1*) because they target the same pathway. Examples of synthetic lethality caused by oncogene activation and tumor-suppressor mutation are much less common, and few (if any) have been validated in vivo.

Here, we show that amplification of the *MYCN* oncogene and inactivation of the *ATRX* tumor-suppressor gene are mutually exclusive in neuroblastomas from patients of all ages and stages of disease. One discrepant tumor sample may have contained two separate clones, but more-detailed analysis was not possible owing to limited tissue. In mouse models and cell lines, the combination of elevated MYCN expression and ATRX loss led to synthetic lethality. Ectopic MYCN caused metabolic reprogramming, mitochondrial dysfunction, ROS production, and DNA damage in *ATRX*-mutant neuroblastoma cells. We propose that *MYCN* amplification and *ATRX* mutations are incompatible in neuroblastoma, because both lead to DNA-replicative stress[10,38,39]. Consistent with this model, the synthetic lethality was partially rescued by genes that reduce oxidative stress (*CUX2*) and pharmacological agents that induce differentiation (retinoic acid) or reduce ROS levels (*N*-acetyl cysteine). Similarly, pharmacological agents that induced replicative stress through DNA damage exacerbated the synthetic lethality.

*ATRX*-mutant neuroblastomas have several unique features, relative to other *ATRX*-mutant cancers. First, *DAXX* was not mutated in our cohort, whereas in pancreatic neuroendocrine tumors (PanNETs), *ATRX,* and *DAXX* mutations have approximately the same frequency and are mutually exclusive[50]. Second, *ATRX*-mutant neuroblastomas have worse outcome, but patients with *ATRX/DAXX*-mutant PanNETs tumors have prolonged survival[50]. Third, *ATRX* mutations in neuroblastomas are often in-frame deletions that remove approximately half of the amino terminus of the protein. In other cancers, the mutations are indels or nonsense mutations[50]. In the original study describing *ATRX* mutations in neuroblastoma[3,16], there was no difference in outcome or clinical presentation for patients with in-frame deletions versus missense or nonsense mutations. However, that cohort was relatively small and a much larger study of *ATRX*-mutant neuroblastomas would be required to determine if there is any genotype–phenotype correlation for the type of *ATRX* mutation.

Previous studies suggest that amino acids 1–841 of ATRX are sufficient for localization to heterochromatin[33]. This region is deleted in the ATRX$^{IFD}$s in most *ATRX*-mutant neuroblastomas. The ADD domain (amino acids 168–293) binds to histone H3 tails, with preferential binding to H3K9me3 domains that lack H3K4 methylation. HP1 can bind H3K9me3 and recruit ATRX through the HP1-interacting domain (amino acids 586–590), even in the absence of the ADD domain in ATRX. Therefore, in neuroblastomas with in-frame deletions, we propose that they lack the

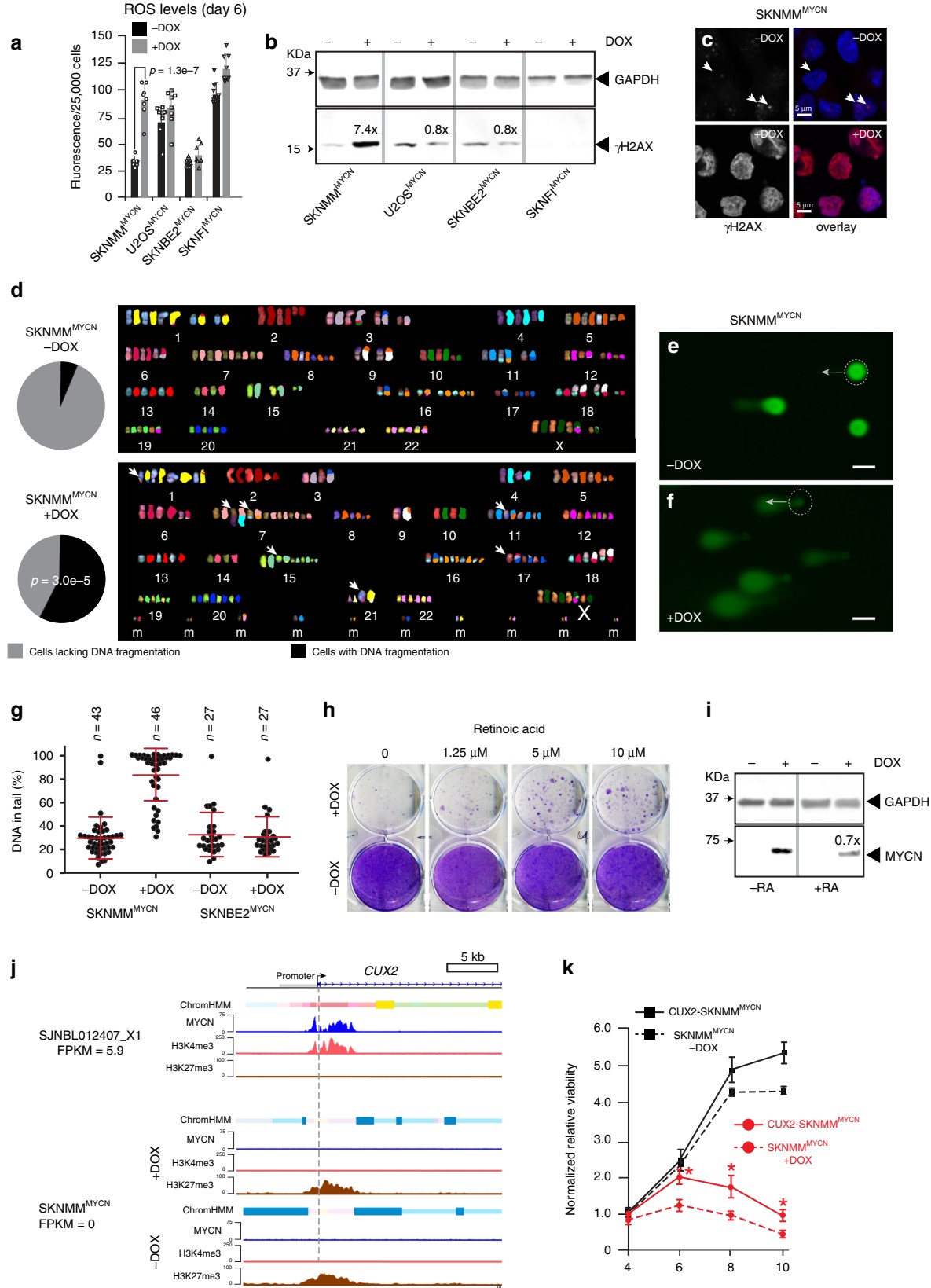

heterochromatin-binding domain and have defects in H3.3-chaperone function that is required for genome maintenance. Our data also suggest that there are defects in H3.3 deposition at promoters and enhancers for genes involved in neuronal differentiation,

particularly for those with G4 structures predicted to form R-loops. Indeed, one of the major pathways that was deregulated is involved in retinoic acid response. This is important because retinoic acid induces differentiation of NB cells and is part of the treatment

**Fig. 5 Expression of MYCN in *ATRX*-mutant cells leads to ROS production, DNA damage, and replicative stress. a** Bar plot (mean±SD) of ROS levels on Day 6 ± doxycycline. *P* value was calculated using two-tailed Student *t* test. **b** Immunoblots on Day 4 ± doxycycline. The experiment was repeated with the same results. **c** Immunofluorescent detection of γH2AX (red) and nuclei (blue) in SKNMM^MYCN^ cells on Day 4 ± doxycycline. The experiment was repeated with the same results. **d** Spectral karyotype analysis (SKY) of SKNMM^MYCN^ cells ± DOX on Day 8. Chromosomes are shown adjacent to the pseudo-colored representation. Arrows indicate translocations. The pie charts show the proportion of cells with DNA fragmentation (*n* = 50). *P* value was calculated using Chi-square test. **e, f** Micrographs of single-cell electrophoresis of individual nuclei (dashed circles) and their COMET tail (arrows) ± doxycycline on Day 5. **g** Mean±SD of the COMET assay scoring. The number of analyzed cells are presented on the graph. **h** Photograph of cresyl violet-stained colonies from SKNMM^MYCN^ cells ± doxycycline, with increasing concentrations of retinoic acid (RA). **i** Immunoblots of SKNMM^MYCN^ cells ± doxycycline, with or without 5 μM RA. The level of MYCN protein was reduced by 30% (0.7×) in the presence of RA. **j** ChromHMM and ChIP-seq for MYCN, H3K4me3, and H3K27me3 for the *CUX2* promoter for a *MYCN*-amplified neuroblastoma (SJNBL012407_X1) xenograft and SKNMM^MYCN^ cells in the presence or absence of doxycycline after 4 days in culture. The gene expression (FPKM) is indicated, and the dashed line indicates the start of transcription. **k** Line plot of SKNMM^MYCN^ cells in the presence or absence of doxycycline after 4, 6, 8, and 10 days in culture with or without ectopic expression of *CUX2* or the *GFP* control from lentiviral infection. Each point is the mean and standard deviation of triplicate experiments, and the asterisks indicate statistical significance (*P* = 0.005, 0.001, and 0.008 at days 6, 8, and 10, respectively, two-tailed Student's *t* test). Scale bars **e, f**, 10 μm. *DOX* doxycycline, *RA* retinoic acid, *ROS* reactive-oxygen species, *m* marker chromosome fragments that could not be definitively identified.

regimen for NB patients. It is possible that there may be a block in retinoic acid response in *ATRX*-mutant NB cells and this may contribute to the poor outcome for those patients.

Based on our data, we propose that *ATRX* mutation contributes to tumorigenesis in two ways. First, defects in H3.3 deposition at telomeres leads to telomere maintenance through ALT. Second, *ATRX* mutations lead to defects in H3.3 deposition at G4 structures in promoters and enhancers of genes involved in neuronal differentiation including retinoic acid-responsive genes. As a result, gene expression is attenuated and *ATRX*-mutant neuroblastoma cells continue to proliferate. Although *DAXX* mutations can also contribute to ALT, they do not alter H3.3 deposition at G4 structures. We propose that both mechanisms (ALT and G4 resolution) are essential for tumorigenesis and that is why we did not detect any *DAXX* mutations in our cohort.

Although the inability of *ATRX*-mutant neuroblastoma cells to resolve G4 structures promotes tumorigenesis by preventing differentiation, it also causes DNA-replicative stress and slows tumor growth. Indeed, it was recently shown that ATRX can play a role in suppressing R-loop (RNA-DNA hybrid) formation[51]. R-loops can lead to collapse of the DNA replication fork and replicative stress[52,53]. This may be why *ATRX*-mutant neuroblastomas are slow growing and indolent. DNA-replicative stress leads to synthetic lethality with ectopic *MYCN* expression. This genetic synthetic lethality may eventually be exploited in the clinic by reducing the function of ATRX in patients with *MYCN*-amplified tumors or inducing metabolic (or other changes) caused by MYCN expression in *ATRX*-mutant tumors. To achieve this ambitious goal in high-risk neuroblastoma, much will need to be learned about the downstream targets of ATRX and MYCN that contribute to this phenotype.

## Methods

**Uncropped blots and flowcytometric analysis gating**. All uncropped gels and blots are shown in Supplementary Fig. 12. Flow cytometric analysis gating information is shown in Supplementary Fig. 13.

**Patient samples**. Patients were eligible for inclusion in the analytic cohort if they enrolled in the COG Neuroblastoma Biology study ANBL00B1 before treatment; had a confirmed diagnosis of neuroblastoma; and had reported outcome data. Informed consent of the patients and/or parents/legal guardians was obtained at the time of enrollment to ANBL00B1. Detailed description of the statistical analysis is provided in the supplemental information.

**Statistical analysis of *ATRX* mutations and clinical features**. The cohort used in this analysis consists of 477 neuroblastoma patients (of which 476 had clinical data and 475 had outcome data available) representing a mixture of risk levels, disease stages, and ages at diagnosis. All are enrolled on COG protocol ANBL12B8 ("Analysis of *ATRX* Mutations in Neuroblastoma") and were assayed for *ATRX*

mutations. Roughly 80% of the patients were diagnosed between 2010 and 2012, with the remaining 20% diagnosed between 2001 and 2009.

The frequency of *ATRX* mutations, along with Clopper–Pearson exact 95% confidence intervals, were calculated across the entire cohort and in subgroups formed by the combination of age categories (12 yrs at diagnosis) and INSS stage categories (1, 2 A/B or 3, 4 S, and 4). Chi-square tests or Fisher's exact tests, depending on sample size, were used to test for associations between *ATRX* mutations and clinical factors. EFS and OS were compared between patients with vs. without *ATRX* mutations, across the entire cohort and in subgroups based on stage, age, and risk. EFS and OS Cox models were used to assess the prognostic strength of *ATRX* mutations in the presence of standard clinical risk factors.

A forward-selection process was used to construct parsimonious Cox models. At the beginning of the forward-selection process, the following candidate variables were considered to be on equal footing: age at diagnosis (< 18 months vs. > 18 months) INSS stage (non-stage 4 vs. stage 4), *MYCN* status (non-amplified vs. amplified), *ALK* mutation, ploidy (hyperdiploid vs. diploid), 1p (no LOH vs. LOH), 11q (no LOH vs. LOH), INPC histology (favorable vs. unfavorable), *ATRX* mutation, mitosis–karyorrhexis index (MKI, low/intermediate vs. high), and grade (totally undifferentiated/poorly differentiated vs. differentiating). Variables were entered into the model one at a time, with the variable chosen for entry being the one that is most significant at each step, per a Wald test. If at any point in the process histology was chosen to enter the model, then MKI, grade, and age at diagnosis were no longer considered for entry, as histology is confounded with these variables. Conversely, if at any point MKI, grade, or age at diagnosis entered the model, then histology was no longer considered for entry. The selection process ended when all remaining candidate variables failed to reach significance at the 0.05 level.

EFS time was measured as the number of days from diagnosis to date of relapse, disease progression, secondary malignancy, death, or, if no event occurred, date of last contact. OS time was measured as the number of days from diagnosis to date of death, or, if the patient did not die, date of last contact. *MYCN* amplification status was assessed both by the neuroblastoma reference lab as part of ANBL00B1 enrollment and by St. Jude Children's Research Hospital. There were some discrepancies between the two, particularly in one patient with an *ATRX* mutation (an aberration which previous research suggests is mutually exclusive from *MYCN* amplification), whom the six neuroblastoma reference lab determined to have *MYCN* amplification, whereas St. Jude detected no amplification. Owing to these discrepancies, both sets of *MYCN* amplification data were considered individually in this analysis.

Outcome was not compared in the <18 months old at diagnosis subgroup, as there were no patients with ATRX mutations in that subgroup. Nor was outcome compared in the 18 mo–5yrs old at diagnosis subgroup, as only one patient in that subgroup had an *ATRX* mutation.

The median follow-up time for patients who did not have an event was 3.9 years. The median follow-up time for patients who did not die was also 3.9 years. The forward-selection process substituting *MYCN* amplification data from Dr. Federico's lab for those from the NBL reference lab yielded the same final EFS and OS Cox models. The effect of *ATRX* mutations on outcome was also tested among patients with INSS stage 4 disease who were at or above 5 years of age at the time of diagnosis. This subgroup was of interest because of its association with a chronic/indolent course of disease. A notable difference between this subgroup and the subgroups in which significant associations between outcome and *ATRX* mutation were observed, was the number of patients with high-risk disease; the subgroup of older INSS stage 4 patients was entirely high risk, whereas the other subgroups included a mixture of risk levels. This prompted us to perform a subgroup analysis based on risk group (high vs. low/intermediate). There were five low/intermediate risk patients with *ATRX* mutations, two of which experienced an event and one subsequently died. The differential effect of *ATRX* mutations between high- and non-high-risk disease groups was also detected in the Cox model of EFS that included an interaction term for mutation and risk level.

Mutual exclusivity between ATRX mutations and MYCN amplification in neuroblastoma was done by merging the COG, TARGET, and PCGP data.

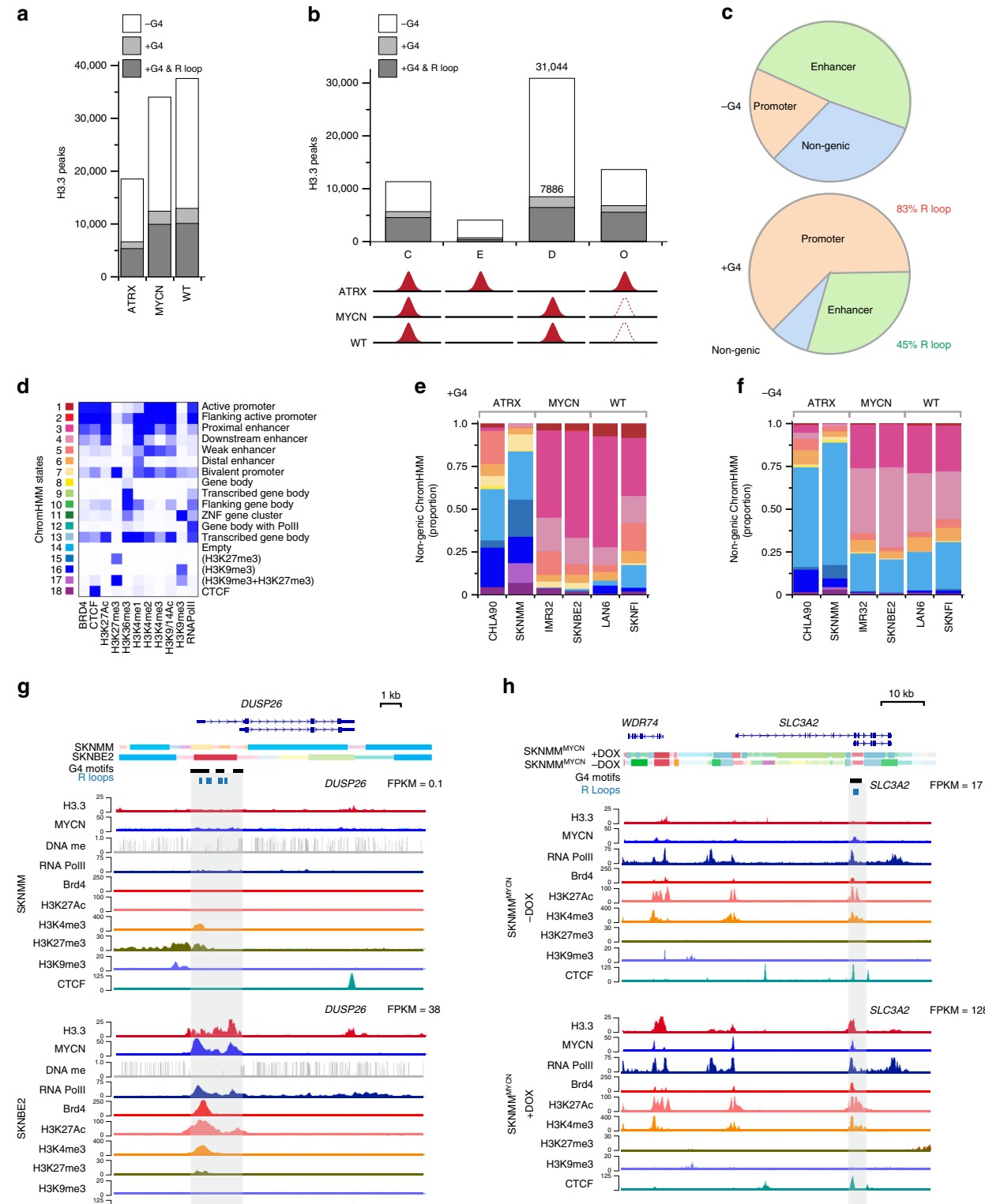

**Custom capture for *ATRX, MYCN* and *ARID1A/B*.** Targeted enrichment was performed using the Seqcap EZchoice Kit (Roche) according to vendor instructions for the Kapa workflow with 500 ng of genomic DNA as the starting input for library construction.

**Exome sequencing.** Whole-exome sequencing was conducted using the SeqCap EZ HGSC VCRome (Roche) according to manufacturers' instructions.

**MYCN FISH.** Purified human *NMYC* BAC DNA (RP11-1183P10) was labeled with a red-dUTP (AF594, Molecular Probes) and human chromosome 2 control

(2q11.2) BAC DNA (RP11-527J8) was labeled with a green-dUTP (AF488, Molecular Probes) both by nick translation. The paraffin slides were deparaffinized with xylene 2 × 10 min each at room temperature (RT), placed in ethyl alcohol 3 × 2 min each at RT, air-dried, placed in 10% buffered formalin for 1 h at RT. The slides were then placed in pepsin (8 mg/ml) in 0.1 N HCL for 3 min, rinsed in dH$_2$O for 5 min at RT. One-hour incubation in Carnoy's fixative at RT was performed. The labeled probes were combined with human sheared DNA and hybridized to the slides in a solution containing 50% formamide, 10% dextran sulfate, and 2× SSC. The probe and slide were co-denatured at 90 °C for 12 min and incubated overnight at 37 °C on a Thermobrite. In brief, washed in PN and then stained with 4,6-diamidino-2- phenylindole (DAPI) (1 µg/ml). Images were captured using a

**Fig. 6 H3.3 deposition is altered at G4 sequences in *ATRX*-mutant neuroblastomas. a** Stack bar plot of H3.3 peaks in the *ATRX*-deficient (CHLA90, SKNMM), *MYCN* amplified (SKNBE2, IMR32), and *ATRX* wild type, *MYCN* non-amplified (LAN6, SKNFI) cell lines. The H3.3 peaks that overlap with G4 sequences are shown in gray. **b** Stack bar plot of H3.3 peaks that are constitutive (C) across all three groups (ATRX, MYCN, WT), enriched (E) in ATRX mutant, depleted (D) in ATRX mutant, or overlap (O) between ATRX mutant and MYCN or WT. The H3.3 peaks that overlap with G4 sequences are shown in gray and the number of peaks for the D group are shown. **c** Piechart of the location of D group H3.3 peaks that have a correlation between the presence or absence of an H3.3 peak and ChromHMM state. Separate pie charts are shown for those that overlap with G4 sequences and those that lack G4 sequences. **d** Heatmap of the ChromHMM states used in this study with color coding. **e** Stack bar plot of the distribution of ChromHMM states for each cell line in the non-genic regions that have D group H3.3 peaks that correlate with ChromHMM state and overlap with G4 sequences or lack **f** G4 sequences. **g** ChromHMM, WGBS, and ChIP-seq tracks for the *DUSP26* gene in SKNMM (*ATRX* mutant) and SKNBE2 (*MYCN* amplified) cells. Gene expression is indicated (FPKM) and G4 motif sequences are shown below the ChromHMM tracks. **h** ChromHMM, WGBS, and ChIP-seq tracks for the *SLC3A2* gene in SKNMM[MYCN] cells with and without doxycycline. Gene expression is indicated (FPKM) and G4 motif sequences are shown below the ChromHMM tracks. *DOX* doxycycline, *ChromHMM* chromatin hidden markov modeling, *G4* guanine quadruplex structure.

---

**Table 2 Numbers of enrolled mice for neuroblastoma genetic mouse model survival study.**

| Strain | Enrolled | Did not reach the survival age | Censored | Reported in the survival analysis |
|---|---|---|---|---|
| *MYCN* | 45 | 8 | 1[#] | 36 |
| *MYCN-ATRX* | 59 | 13 | 2[##] | 44 |
| *MYCN-ALK* | 18 | 1 | 0 | 17 |
| *MYCN-ALK-ATRX* | 47 | 12 | 8[*] | 27 |

[#]Found dead no tumors.
[##]Hunched and lethargic no tumors, large spleen.
[*]Malocclusion, four mice with fight wounds, paralysis hind limbs, found dead no tumors, large thymus (dyspnea).

Plan-Apochromat ×63 objective on a Zeiss Axio Imager.Z2 microscope and GenASIs scanner (ScanView software version 7.2.7).

**ATRX IHC.** ATRX immunohistochemistry was done on formalin-fixed 4-μm-thick paraffin sections using Leica Polymer Refine Detection kit (Leica Microsystems, Wetzlar, Germany) on a Leica Bond system. Epitope retrieval was done by heating and anti-ATRX antibodies polyclonal antibodies (1:600; Sigma-Aldrich) was used and incubated for 15 min[3].

**Histopathologic review of genetically engineered mouse models.** Adrenals and paravertebral sympathetic ganglia from *Atrx^flox*-LSL-MYCN:Dbh-iCre mice at the age of three weeks and 1 year by hematoxylin and eosin (HE) staining and immunohistochemical staining for Ki-67, cleaved caspase 3, tyrosine hydroxylase, and synaptophysin. Formalin-fixed adrenal glands and sympathetic ganglia were processed and embedded in paraffin by standard techniques, sectioned at 4 μm, mounted on positively charged glass slides (Superfrost Plus; Thermo Fisher Scientific, Waltham, MA), and dried at 60 °C for 20 min. Tissue sections were then stained with H&E or subjected to immunohistochemical (IHC) staining. For IHC, tissue sections underwent antigen retrieval in a prediluted Cell Conditioning Solution (CC1; Cat# 950-124; Ventana Medical Systems, Indianapolis, IN) for 32 min on a Discovery Ultra immunostainer (Ventana Medical Systems, Tucson, AZ) in order to detect tyrosine hydroxylase, synaptophysin, or Ki-67 antigen. The primary antibodies used in this study included the following: A rabbit polyclonal antibody raised against tyrosine hydroxylase diluted 1:500 (cat# AB152, EMD Millipore, Billerica, MA); a rabbit polyclonal antibody to synaptophysin diluted 1:400 (cat# ab7837; Abcam, Cambridge, UK); and a rabbit monoclonal antibody to Ki-67 antigen (Clone SP6) diluted 1:50 (cat# RM-9106-S; Thermo Scientific, Fremont, CA). Binding of these primary antibodies was detected using OmniMap anti-Rabbit (#760-4311; Ventana Medical Systems), with DISCOVERY Chromo-Map DAB Kit (Ventana Medical Systems) as chromogenic substrate. Stained sections were examined by a pathologist blinded to the experimental group assignments.

**Orthotopic patient-derived xenografts.** Mouse studies were performed in a strict accordance with the recommendations in the Guide to Care and Use of Laboratory Animals of the National Institutes of Health. The protocol was approved by the Institutional Animal Care and Use Committee (IACUC) at St. Jude Children's Research Hospital. All efforts were made to minimize suffering. All mice were housed in accordance with approved IACUC protocols. Animals were housed on a 12–12 light cycle (light on 6:00, off 18:00) and provided food and water ad libitum.

Patient-derived xenograft cells were generated and maintained orthotopically in athymic nude (Jackson Laboratories, strain code 007850), NSG (Jackson Laboratories, strain code 005557) or C57BL/6 scid mice (Jackson Laboratories, strain code 001913) as described previously[54]. Tumor cells were dissociated using

Trypsin (10 mg/ml, Sigma Cat # T9935) filtered and then suspended in a Matrigel basement membrane matrix (BD Biosicences catalog number 354234) at a concentration of $2 \times 10^5$ cells per 10-μl and placed on ice for injection. The cells were injected in the para-adrenal region of the mice with ultrasound guidance under anesthesia.

**Genetic mouse models.** To test the incompatibility between *ATRX* mutations and *MYCN* overexpression in mice, we used *LSL-MYCN:DbhiCre* mice[17]. These mice express *MYCN* from the *Rosa26* locus when a stop sequence is floxed out by Cre recombinase under the control of *Dbh* promoter in sympathetic ganglion cells. We breed *LSL-MYCN:DbhiCre* mice to *Atrx^Flox* mice[8] to inactivate *Atrx* in the same tissues. We also tested the consequences of *Atrx* inactivation when *Th-ALKF^1174L* allele is added[18] (*LSL-MYCN: DbhiCre:Th-ALK^1174L*). *Th-ALK^F1174L* mice constitutively express *ALK^1174L*, the most common activating *ALK* mutation found in human neuroblastoma, under the control of *Th* promoter.

Mice were genotyped using the published protocols[8,17,18]. Mice with the desired genotypes were enrolled in the study at the time of weaning and were followed up by abdominal palpation at least once every 2 weeks starting from age of 6 weeks (Table 2). Enrolled mice that died with no detectable tumors were censored out of the survival study. Mice were killed when the tumor is 20% of body weight or when the mouse has other morbidities, e.g., lethargic, hunched or dehydrated. After euthanasia, mice were opened longitudinally to expose abdomen, thorax, and neck to search for tumors. Tumors were dissected out and were cut into pieces for histological, immunohistochemical, DNA, RNA, and protein preparations.

**Whole-genome sequencing.** DNA from the tumors and matching germline samples was sequenced and analyzed as described previously[54]. In brief, 250–500 ng of genomic DNA was input for library construction using Illumina-compatible adapters, and four to six cycles of amplification were performed with KAPA HiFi Hotstart ReadyMix (KAPA Biosystems). The reference human genome assembly NCBI Build 37 was used for mapping samples. WGS mapping, coverage and quality assessment, single-nucleotide variation (SNV)/indel detection, tier annotation for sequence mutations, and identification of LOH have been described previously[55]. Structural variations were analyzed and annotated using CREST[56]. Copy number variations were identified by evaluating the difference of read depth for each tumor and its matching normal using CONSERTING[57]. WGS data were uploaded at EMBL with accession number EGAS00001002528.

**CUT&RUN and analysis.** We did automated CUT&RUN (AutoCUT&RUN) for H3.3 as described in detail https://www.protocols.io/view/autocut-run-genome-wide-profiling-of-chromatin-pro-ufeetje. In brief, cells were bound to concanavalin A-coated magnetic beads (Bangs Laboratories), permeabilized with digitonin and then incubated with primary antibody against H3.3 (Abnova, clone 2D7-H1), followed by incubation with rabbit anti-mouse IgG (Abcam, ab46540). Cells were arrayed on a 96-well plate and the rest of the reaction was carried out on a Beckman Biomek FX, including digestion with proteinA-MNase, ligation of adapters and library preparation. Size distributions of prepared libraries were assessed using an Agilent 4200 TapeStation. Libraries were pooled and at equimolar concentrations and 25 × 25 bp paired-end sequencing on an Illumina HiSeq 2500 was performed.

To find differential binding sites for H3.3, for each CUT&RUN sample, we called peaks twice by MACS2 (paired-end mode) that one using default FDR corrected $p$ value cutoff 0.05 as high confidence peak set and the other using 0.5 as low confidence peak set. Then for each of the three groups, we compiled the reproducible peak set that required a high confidence peak also overlap a low confidence peak from the other samples within the sample group. Then we merged the reproducible peaks from three group to create a reference peak set and counted fragments (fragment size <2000 bp) overlap the reference peaks for each sample. At last, we performed statically tests using Voom[58]. To check their overlapping with G4 motifs and R-Loops, we downloaded G4 Motifs from supplementary data of ref. [59] and lifted over them from hg18 to hg19, we got 100,780 regions. We

downloaded 169,222 "RLFS in genic and proximal regions" from http://rloop.bii.a-star.edu.sg/ for hg19 as predicted R-Loops regions[60].

**Whole-genome bisulfite sequencing and analysis.** Genomic DNA was extracted using the DNeasy Kit (QIAGEN #69504) according to the manufacturer's protocol and quantified with a Nanodrop (Thermoscientific). Whole-genomic bisulfite conversion and sequencing were done by Hudson Alpha (Huntsville, AL).

Sequencing data were aligned to the hg19 human genome using BSMAP2.74[61]. We first performed statistical test of differentially methylated locus (DML) between MYCN-amplified and non-MYCN-amplified samples using DMLtest function (smoothing = TRUE) in DSS (Wu et al. 2015), the results were then used to detect differentially methylated regions (DMRs) using CallDMR function in DSS, the p value threshold for calling DMR is 0.01. The minimum length for DMR is 50 bps and the minimum number of CpG sites for DMR is 3. The minimum methylation difference is 0.2. The DMRs that overlap with the promoters (1 kb flanking the TSS) of differentially expressed genes (at least twofold changes and adjust p value ≤ 0.1) were deemed to be associated with gene expressions.

**RNA-Seq and analysis.** RNA was extracted from cultured cells and tissues using RNeasy Kit (QIAGEN #74104) or TRIzol (Life Technologies) preparations via a phenol–chloroform extraction or Direct-zol RNA MiniPrep (Zymo Research # R2052) following manufacturers' protocols. RNA concentration was measured using a NanoDrop (Thermo Fisher scientific, Waltham, MA) and the quality of RNA was determined with a bioanalyzer (Agilent Technologies, Santa Clara, CA). Libraries were prepared using the TruSeq Stranded Total RNA Library Prep Kit (Illumina, San Diego, CA) from 500-ng total RNA. Paired-end deep sequencing was done using HiSeq 2500 sequencers (Illumina, San Diego, CA).

FASTQ sequences were mapped to the human genome hg19 (GRCh37-lite) and Gene-based FPKM quantification was calculated similar to previously described[57] using GENCODE gene annotation (v26lift37). We then used VOOM[58] for differential analysis between +DOX vs −DOX. To confirm the results were meaningful, we ranked the genes by log2 fold change and use GSEA[62] to check if any MYCN-related gene sets in MSigDB(v6.1) could be found enriched. Indeed, WEI_MYCN_TARGETS_WITH_E_BOX[63] gene set were found top enriched for upregulated genes in +DOX (rank 46 out of 11371 gene sets, NES = 3.01, FDR corrected p value < 1e-4).

**ChIP-Seq.** Antibodies used in this study to generate ChromMM were previously validated and the validation data and protocols are available through St. Jude Childhood Solid Tumor Network website (https://www.stjude.org/CSTN/)[20].

For MYCN ChIP-Seq, we used rabbit polyclonal antibody (Active Motif Cat# 61185). This antibody was validated by small-scale and large-scale sequencing by active motif as well as by us in St. Jude Children's Research Hospital. We tried different amounts of the antibodies and chromatin and determined that 20-μg of the antibody and 200-μg of the chromatin per ChIP-Seq reaction show the best results.

Chromatin in cultured cells was fixed by changing the medium with fresh medium containing 1% formaldehyde (Thermo scientific # 28906). Snap-freezed xenografts were pulverized and hammered to powder in liquid Nitrogen, suspended in PBS and fixed by adding formaldehyde (Thermo scientific # 28906) to final concentration of 1%. Fixation was done for 10 min at room temperature. Cross-linking was stopped by adding glycine to a final concentration of 1.25 M. Cells and xenografts were washed in 1× PBS containing 1× protease inhibitor (Pierce # 78430).

Chromatin preparation and shearing were done using TruCHIP chromatin shearing kit (Covaris #520127) following the manufacturer's protocol. Chromatin immune-precipitation (ChIP) reactions were performed using iDeal ChIP-seq kit (Diagenode # C01010051) with 10% of chromatin from each reaction was used as an input. Precipitated DNA was de-crosslinked, extracted using MinElute PCR-purification kit (QIAGEN #28006) and quantified using the Quant-iT PicoGreen ds DNA assay (Life Technologies #Q33120). Quantitative polymerase chain reaction (qPCR) was done to assess the efficiency of ChIP reactions as described in the validation protocols. Five to ten nanograms of the input and precipitated DNA were used to prepare the sequencing libraries using the NEBNext ChIP-Seq Library Prep Reagent Set for Illumina with NEBNext High-Fidelity 2× PCR Master Mix following the manufacturer's instructions (New England Biolabs) with some modificaiton as described previously;[64] after adaptor ligation, 1:1 Ampure cleanup was added twice, no Ampure size-selection step and the extension was done for 45 seconds. Inset-size distribution of the completed libraries was analyzed using a 2100 BioAnalyzer High Sensitivity kit (Agilent Technologies) or Caliper LabChip GX DNA High Sensitivity Reagent Kit (PerkinElmer). ChIP reactions and library preparation for RNA-Pol II, Brd4 H3K9me3 and MYCN were done by Active Motif. Quant-iT PicoGreen ds DNA assay (Life Technologies) and Kapa Library Quantificaiton kit (Kapa Biosystems) or low-pass sequencing on a MiSeq nano kit (Illumina) were used to quantify the libraries. Fifty-cycle single-end sequencing was performed on an Illumlina HiSeq 2500.

**ChIP-Seq analysis.** ChIP-Seq analysis was done as described previously[64]. In brief, ChIP-Seq reads were aligned to human genome hg19 (GRCh37-lite) using BWA

software (version 0.7.12-r1039, default parameter) and the duplicated reads were marked using Picard software. We kept only non-duplicated reads for the analysis using samtools (parameter "-q 1 -F 1024" version 1.2). The quality control of the data followed ENCODE criteria. We calculated relative strand correlation value (RSC) and estimated the fragment size under support of R (version 2.14.0) with packages caTools (version 1.17) and bitops (version 1.0-6). We required at least 10 M unique reads for point source factors H3K4me2/3, H3K9/14Ac, H3K27Ac, CTCF, RNA PolII, BRD4 with RSC > 1, 20 M unique reads for broad markers (H3K4me1, H3K9me3, H3K27me3, H3K36me3) and 10 M unique mapped reads for INPUTs with RSC < 1. All samples were manually inspected and the SPP (version 1.1) was used to generate the cross-correlation plot. Then we generated bigwig files from the best fragment size (the smallest fragment size estimated by SPP). Bigwig files were examined using IGV genome browser for clear peaks and low background noise. MACS2 (version 2.0.10.20131216)[65] were used to call peaks for MYCN ChIP-Seq data with FDR corrected p value 0.05 (−nomodel −extsize FRAGMENTSIZE, where FRAGMENTSIZE was estimated by SPP as described above). Genes promoter (TSS ±2 kb) with MYCN peaks called were considering have MYCN-binding site. Enrichr[66] were used for pathway analysis.

**Super-enhancer and core regulator circuit analysis.** Super-enhancers and core regulator circuit analysis have been performed as described previously[64]. We called auto-regulators by CRCMapper[67] based on super-enhancers called by ROSE[68]. We provided two sets of results that one only used H3K27ac data, while the other one excluded H3K27ac peaks overlapping promoter (defined by H3K4me3 peaks) before calling super-enhancer by ROSE.

**ChromHMM.** ChromHMM models were generated for y as described before[54]. In brief, Non-duplicated aligned reads were extended by fragment size, and ChromHMM (version 1.10, with '-colfields 0,1,2,5 –center' for BinarizeBed) was used for chromatin state modeling. To choose the state number, we first modeled all samples together from seven states to 33 states and selected the model with 18 states upon manual inspection. For better visualization of the dynamics of HMM state across stages, we normalized color intensity by the maximum total percentage of a state covering a gene and flanking region. We reduced the interval for an HMM state to half bar and the intensity to half the normalized amount if it did not rank in the top two HMM state for a gene. As HMM states could be assigned by multiple genes, the maximum total percentage across genes was used for normalization.

We applied the Hidden Markov Model trained for 18 states from previously four types of solid tumors to the data from this manuscript. Expanded chromHMM state 13 were defined as proportion of promoter downstream (TSS ~ TSS + 2 kb) in SKNMM^MYCN + Dox > twofold of SKNMM^MYCN − DOX and ≥ 20% in SKNMM^MYCN + DOX.

**Immunoblotting.** The medium was removed and the cells were washed twice with PBS then the cells were scrapped in 1-ml PBS and centrifuged at 2000 × g for 5 min at 4 °C to pellet the cells. PBS was removed and the cells were flash-freezed in dry ice and stored at −80 °C until use. Cell pellets were thawed on ice and lysed for 15 min in radioimmunoprecipitation assay buffer (Sigma Cat. #R0278) supplemented with 1 × Halt protease inhibitor (Thermo Fischer, Cat. #7834) and 5 mM ethyle-nediaminetetraacetic acid (EDTA). Lysate were then centrifuged for 15 min at 15,000 rpm at 4 °C. Protein concentration in the cell lysates was measured using BCA protein assay kit (Thermo Fischer, Cat. #232225). Equal amounts of total protein were resolved on 4–15% sodium dodecyl sulfate polyacrylamide gel (Biorad Cat#4561086) and transferred to a nitrocellulose membrane by electrophoresis at 30 volts overnight at 4 °C. Non-specific binding was blocked by incubating the membrane with Odyssey Blocking buffer, PBS (LiCore Cat# 927-40000) for 1 h at room temperature. Then the membrane was incubated with the primary antibody overnight at 4 °C. The following antibodies were used: rabbit anti-ATRX (1:1000, Abcam, ab97508), mouse anti-MYCN (1:1000, Santa cruz Biotechnology, sc-53993), rabbit anti-GAPDH (1:2500, Abcam ab-9485), rabbit anti-DAXX (1:1000, Santa cruz Biotechnology, sc-7152) and mouse anti-β-actin (1:5000, Sigma, A1978). Membranes were washed three times in PBS with 0.1% Tween-20 at room temperature then were incubated in the secondary antibodies: IRDye 800CW goat anti-mouse (1:5000, LiCore, 925-68070) or IRDye 680CW goat anti-rabbit (1:5000, LiCore, 925-32211) for 30 min at room temperature. The membranes were then washed three times in phosphate-buffered saline with Tween; 5 min each at room temperatures and one time in PBS. The membranes were then scanned by Odyssey CLX infrared gel imaging system (Li-Cor Biosciences, Lincoln, NE). Band intensities were quantified using ImageStudio Lite software (Li-Cor Biosciences, Lincoln, NE).

**Telomere FISH.** After a 5-hour colcemid incubation, cells from both samples were harvested using routine cytogenetic methods. A commercially prepared directly fluorescein isothiocyanate-labeled PNA telomere probe (DAKO, Cat. # 5327) was used for this analysis. DAKO protocols were followed for the pre-treatment and hybridization steps. The slides were then stained with DAPI. In total, 255 inter-phase nuclei from each sample were individually analyzed for the number of

telomeres/cell, fluorescence/cell, and fluorescence/telomere. A fixed exposure time is used for all cells.

**Telomere qPCR**. Telomere qPCR was done as described previously (Cheung et al.[3]). DNA was extracted using DNAeasy Kit (Qiagen, Cat. # 69504). qPCR was done on 10-ng DNA template using SYBR green Select master mix (Thermo Fisher, Cat. #4472908). Two sets of primer pairs were used in separate reactions were used to amplify the telomeric sequence and a common locus RPLP0, which was used as an internal control. The reactions were done in duplicates and the average delta $C$ (t) was calculated for every sample by subtracting the average $C$(t) values of the telomeric reaction from those of the internal control reactions. The primers used are

Telomere F: 5′-CGGTTTGTTTGGGTTTGGGTTTGGGTTTGGGTTTGGG TT-3′
Telomere R: 5′-GGCTTGCCTTACCCTTACCCTTACCCTTACCCTTACCC-3′
Control, locus:
RPL0 F: 5′-CAGCAAGTGGGAAGGTGTAATCC-3′
RPL0 R: 5′-CCCATTCTATCATCAACGGGTACAA-3′

**ATRX shRNAs**. Eight ATRX-specific shRNA lentiviral vectors were purchased from Dharmacon as bacterial glycerol stocks (Cat. # V2LHS-092920 (#9), V2LHS-202920 (#20), V2LHS-122887 (#22), V3LHS-375686 (#37), V2THS-22887 (#87), V2THS-93289 (#30), V2THS-255090 (#13), and V2THS-201895 (#95)). Lentiviruses were made in HEK293T cells by co-transfecting the viral vectors with three packaging plasmids. The supernatants containing the viral particles after 48 and 72 h were collected, filtered, concentrated using ultracentrifugation, and titrated in HeLa cells using the GFP or RFP reporters in the vector. In addition, two non-silencing shRNA lentiviral particles were used as controls (Cat. # RHS4743 and RHS4346). Cells were transduced with equal number of viral particles and transduced cells were selected with flow cytometry or pouromycin after 48 h post transduction.

**Cell cycle analysis**. Cells were trypsinized and harvested, pelleted and suspended in 1-ml propidium iodide solution (0.05 mg/ml propidium iodide (Sigma, Cat. # P4864), 0.1% sodium citrate, 0.1% Triton X-100). The cells were then treated with 10-ml 0.2 mg/ml of ribonuclease A (Colbiochem, Cat. #556746) for 30 min at room temperature, filtered through 40-μm nylon mesh, and analyzed by flow cytometry for the DNA content.

**Colony assays**. Cells were transduced with a lentivirus expressing ATRX shRNA #9, ATRX shRNA #20, or scrambled control shRNA at multiplicity of infection 5 for 24 h. Then the cells were harvested, suspended into single cells in a pre-warmed medium, counted and seeded at 300 cells/well in six-well plates. The plates were shaken gently every 1–2 min for 10 min. Fresh medium was added every 4 days. After 2–3 weeks when colonies were visible, cells were washed, fixed in 1% paraformaldehyde for 30 min at room temperature and stained with 0.1% crystal violet.

**CRISPR/Cas-9 targeting ATRX and DAXX**. Guide RNAs (gRNAs) were designed targeting an early and conserved exon between isoforms of the hATRX and hDAXX genes. Bioinformatics analysis was performed to identify gRNAs with at least 3 bp of mismatch to other sites in the human genome. gRNAs were further validated in human K562 cells for cutting efficiency using targeted deep sequencing.

Two validated hATRX (g2 and g5) and one hDAXX (g11) gRNAs were used in this study. The sequences of gRNAs are:

hATRX.g2: 5′-CTGCACTGCTTGTGGACAACNGG-3′
hATRX.g5: 5′-CAATGAAGGGTGTCTATAAANGG-3′
hDAXX.g11: 5′-GGTTCTTCTGACAGTAACGANGG-3′
Cells were transiently transfected with PC526-Cas-9 plasmid (to express Cas-9; Addgene # 43945), either ZZ52.hATRX.g2 or ZZ52.hATRX.g5 (to express hATRX gRNA; the backbone plasmid Addgene # 43860) and pUB-GFP (to express GFP; Addgene # 11155) in a ratio of 7:7:1, respectively using Mirus 2020 (Mirus Biotechnology Cat. # 2020). After 48 h, the cells were sorted for GFP-positive cells. Half of the sorted cells were pelleted immediately and stored at −80 °C and the remaining cells were seeded in six-well plates and harvested at different time points. To test for the NHEJ efficiency after CRISPR targeting in IMR32 cells, we transfected the cells with the same mix of plasmids but added a control plasmid expressing gRNA targeting the hAAVS1 locus (Addgene # 41818) in a ratio of 4:4:6:1 for hAAVS1.gRNA: ATRX.gRNA: Cas-9: GFP expressing-plasmids, respectively. DNA was extracted from the cell pellets using DNAeasy kit (Qiagen, Cat. # 69504) and PCR was done to amplify the targeted region using primers to which MiSeq adaptors were annexed and the gel-purified PCR products were subjected to deep sequencing.

The primers used are (adaptor sequences are underlines)

hATRX F: <u>TCGTCGGCAGCGTCAGATGTGTATAAGAGACAG</u>TGGAATTT GCCAAGGT<u>TGTCATGTGC</u>
hATRX R: <u>GTCTCGTGGGCTCGGAGATGTGTATAAGAGACAG</u>CTTTCTT CAAGACTGT<u>GCCCTCAAAGGCC</u>

hAAVS1 F: <u>TCGTCGGCAGCGTCAGATGTGTATAAGAGACAG</u>GTGTCCC CGAGCTGGG<u>ACCA</u>
hAAVS1 R: <u>GTCTCGTGGGCTCGGAGATGTGTATAAGAGACAG</u>TAGGAA GGAGGAGG<u>CCTAAGGATGGG</u>

The percentage of indels was determined using CRISPResso software as described before[69].

**Comet assay**. Alkaline comet assay was done using CometAssay Reagent Kit (Trivegen, Cat. # 4252-40-K) following the manufacturer's protocol. In brief, cells were harvested and suspended in PBS buffer at a concentration of $1 \times 10^5$ cells/ml. Cells were combined with LMAgarose in a ratio 1:10 at 37 °C and 50-μl of the mix was spread over the CometSlide. The doxycycline-treated and untreated samples were loaded onto two different wells on the same slide. Slides were left at 4 °C for 30 min in the dark to allow agarose to polymerize. The slides were then incubated in a freshly made Alkaline Unwinding Solution (200 mM NaOH, 1 mM EDTA, pH > 13) overnight at 4 °C in the dark. Electrophoresis was done at 4 °C for 30 min at 21 volts in the Alkaline Electrophoresis Solution (200 mM NaOH, 1 mM EDTA, pH > 13). The slides were washed twice in dH₂O, dried at 37 °C and stored in the dark. To analyze the samples, nuclei suspended in the agarose gel were stained by 100-μl of SYBR Gold (Thermo Fischer, Cat. #S 11494) diluted 1:30,000 in TE buffer (10 mM Tris-HCL pH 7.5, 1 mM EDTA). Then the slides were rinsed in water and dried completely at 37 °C. Slides were examined by a fluorescence microscopy for excitation/ emission: 496/522 and 10–15 images were taken per sample. Percentage of DNA in the tail was quantified in the images using ImagJ software by measuring fluorescence intensity in the tail/total fluorescence × 100. Mann–Whitney non-parametric statistical test was used to compare the scores in the samples with or without doxycycline.

**Spectral karyotyping (SKY) analysis**. Cells were treated with colcemid for 5 h at 37 °C then the cells were harvested for cytogenetic analysis. SKY analysis was done using a commercially available probe from spectral imaging (Carlsbad, CA) following the manufacturer's protocol. Fifty metaphase cells from every sample were scored for the presence of DNA fragmentations. Statistical analysis was done using Fischer exact test.

**Hydroxyurea sensitivity**. The cells were seeded in 96-well plates at a density of 2500 cells per well over night. Next day (day 0), the medium was changed with a fresh medium with or without 2 μg/ml doxycycline. After 24 h of doxycycline treatment (day 1), the medium was changed again with fresh medium containing serially diluted hydroxyurea with or without doxycycline. At day 4, the cell viability was assessed using the CellTitre-Glo cell viability assay kit (Promega, Cat. # G7572) using PHERAstar FSX (BMG Labtech, San Diego, California).

**Inducible MYCN**. To make the doxycycline-inducible cell lines, we used pLenti4/ TO/V5 DEST system (Invitrogene, Cat. # V480-20) following the manufacturer's protocol. In brief, cells were transduced with the pLenti4/TO lentivirus to constitutively express the Tet repressor and the cells were selected with 10-μg/ml blasticidin-HCL (Thermo Fisher, R21001) in the medium. The cells were then transduced with the V5 DEST-MYCN or V5 DEST-control lentivirus and were selected with 10-μg/ml zeocin (Thermo Fisher, R25001) in the medium. The V5 DEST-MYCN virus was a gift from the Freeman lab at St. Jude Children's Research Hospital. The cells were maintained in a selective medium containing 5-μg/ml blasticidin-HCL and 5-μg/ml zeocin. We also cloned MYCN into another dox-inducible system using Lenti-X Tet-One (Takara Cat# 631847) following manufacturer protocol. We confirmed the sequence of the construct by sanger sequencing. Cells were transduced with Lenti-X Tet-One lentiviruses containing MYCN or with the empty vector. Selection for the cells containing the construct was done using puromycin and the cells were maintained in puromycin-containing media. Induction of MYCN expression was done by adding doxycycline to the medium to a final concentration of 1–2 μg/ml.

**Growth curve**. The cells were plated in six-well plates at a density of 50,000 cells per well over night. Then the medium over the cells was changed with a fresh medium with or without 1–2-μg/ml doxycycline. Three wells from each of the cell lines were harvested every day and counted for the total number of cells per well using a hemocytometer.

**Live imaging**. Live cell time-lapse imaging experiments for SKNMM^MYCN in the presence or absence of doxycycline were performed using a Nikon C2 confocal configured on a TE2000E2 microscope with a 20 × 0.8 NA Plan Apo lens (Nikon Instruments Inc., Melville, NY). A 640 nm DPSS lasers was used to simultaneously excite Annexin V-Alexa Fluor 647 and generate the DIC images. During imaging experiments, the cells were maintained at 37 °C and 5% CO₂, and images were collected every 30 min for 67 h. Images were acquired and processed using NIS Elements software.

**In vivo mixing experiment**. SKNMM^MYCN and SKNMM^CONT cells were labeled with luciferase-YFP (yellow fluorescent protein) using a lentivirus. Two mixed cell

pools were prepared in a 1:1 ratio: labeled SKNMM[MYCN] cells/unlabeled SKNMM[CONT] and labeled SKNMM[CONT] /unlabeled SKNMM[MYCN]. Cell mixes were injected in the para-adrenal region of the mice (four million cells/mouse) with ultrasound guidance under anesthesia as explained before. In this experiment, we used NOD.CB17-Prkdc[scid]/J female mice (Jackson laboratories, Stock # 001303). The mice were followed up for the tumor formation using ultrasound scanning every 2 weeks, xenogen signal every week and palpation every week. Mice were enrolled in the study when the tumor was at least 14 mm$^3$ in size using the ultrasound scanning. Enrolled mice were randomly assigned to receive either doxycycline (2 mg/ml) in the drinking water or continue on regular water.

The labeled/unlabeled cell mixes were also maintained in culture and the ratio of YFP-expressing cells in each cell mix was determined weekly using flow cytometric analysis.

**Lactate measurements**. Lactic acid was measured using Lactate Assay Kit (Sigma Cat. # MAK064) following the manufacturer's protocol. The absorbance at 570 nm (A570) was measured for a colorimetric detection of the lactate using the microplate reader SpectraMax-M5 (Molecular Devices, Sunnyvale, California).

**GC-MS-mediated analysis of 13C-labeled TCA metabolites**. The analytical procedure applied was based on the one previously published[28]. In brief, cells were cultured in glutamine-free medium (10% fetal bovine serum (FBS), penicillin/streptomycin) supplemented with 4 mM of standard glutamine or [U-$^{13}$C]-glutamine (CLM-1822-H); Cambridge Isotope Laboratories, Inc. for 5 h. Next, cells were trypsinized, washed with ice-cold saline solution, snap-frozen, and stored in liquid nitrogen for further processing. Extraction of hydrophilic metabolites was performed using two volumes of methanol/water solution (2:1 v/v, high-performance liquid chromatography (HPLC)-grade), spiked with 5 nM of norvaline as internal standard), homogenization and mixing with one volume of chloroform (HPLC-grade). Derivatization with 2% methoxylamine-HCl solution and GC-MS-grade MTBSTFA + 1% t-BDMCS reagent was performed as previously described. Samples were analyzed in split-less mode using Agilent 7890B GC system equipped with a 5977B MSD, Agilent VF5ms, + 1 m EZ, 60 m, 0.25, 0.25 μM column, helium carrier gas and Mass Hunter software package. Chromatographic gradient conditions: separation time 36 min; start at 80 °C, 1 min; ramp to 250 °C at 7oC/min; ramp to 300 °C at 50 °C/min and hold for 9 min. MSD settings: source 150 °C; quadrupole 150 °C; interface 300 °C; injector 250 °C; EI source 70 eV, EI pressure $1.8 \times 10^{-5}$ torr. Signals were acquired using SIM mode conditions as described before (5) as well as using scanning mode (full scan range 50–500 m/z at ~ 2 scans/s ($N = 3$)). Enrichment of C13-labeled isotopomers of analyzed compounds was corrected by natural abundance using "standard glutamine" medium-incubated sample as reference.

**Detecting metabolites using CE-TOFMS**. MYCN-induced and un-induced SKNMM[MYCN] cells were grown on either glutamine-free or glucose free completed medium supplemented with [U-$^{13}$C]-glutamine (CLM-1822-H) or [U-$^{13}$C] glucose (CLM-1396); Cambridge Isotope Laboratories, Inc. for 18 h. Metabolites were extracted from the cells following the protocol provided by Human Metabolome Technology Inc, (Boston, MA). Samples were filtered through 5-kDa cutoff filter (ULTRAFREE-MC-PLHCC, Human Metabolome Technologies, Yamagata, Japan) and sent to Human Metabolome Technologies Inc. for analysis. We also sent snap-freezed medium from the same samples for metabolomic analysis.

Metabolites were measured in both Cation and Anion modes using Agilent Capillary electrophoresis–time of flight mass spectrometry (CE-TOFMS) system (Agilent Technologies Inc.) and Capillary: Fused silica capillary i.d. 50 μm Å ~ 80 cm with the following conditions:

Cationic metabolites (cation mode):
Run buffer: cation buffer solution (p/n: H3301-1001)
Rinse buffer: cation buffer solution (p/n: H3301-1001)
Sample injection: pressure injection 50 mbar, 10 sec
CE voltage: positive, 27 kV
MS ionization: ESI positive
MS capillary voltage: 4000 V
MS scan range: m/z 50–1000
Sheath liquid: HMT sheath liquid (p/n: H3301-1020)
Anionic metabolites (anion mode):
Run buffer: anion buffer solution (p/n: H3302-1021)
Rinse buffer: anion buffer solution (p/n: H3302-1021)
Sample injection: pressure injection 50 mbar, 25 sec
CE voltage: positive, 30 kV
MS ionization: ESI negative
MS capillary voltage: 3500 V
MS scan range: m/z 50–1000
Sheath liquid: HMT sheath liquid (p/n: H3301-1020)
Data processing and analysis

Data processing and analysis were done by Human Metabolome Technologies. In brief, MasterHands ver. 2.17.1.11 software was used to extract peaks detected in CE-TOFMS analysis in order to obtain m/z, migration time (MT), and peak area. Relative peak area was calculated as:

Relative peak area = metabolite peak area/(internal standard peak area × sample amount)

Signal-noise ratio (S/N) = 3 was set as a threshold for peak detection limit.

Human Metabolome Technologies target library was used to identify putative metabolites and their isotopic ions on the basis of m/z and MT with tolerance ± 0.5 min in MT.

Absolute quantification was performed in total amount of each detected metabolite. All the metabolite concentrations were calculated by normalizing the peak area of each metabolite with respect to the area of the internal standard and by using standard curves, which were obtained by single-point (100 μM) calibrations.

**pH measurements**. Five milliliters of the medium were centrifuged at $500 \times g$ for 5 min to remove suspended cells and the pH was measured using a pH meter (Mettler Toledo, Colombus, OH).

**Mitotracker green and TMRE assays**. MitoTracker Green FM (Thermofisher # M7514) or Tetramethylrhodamine ethyl ester perchlorate (TMRE, Thermofisher # T669) was added directly to the medium to a final concentration of 200 nM or 50 nM, respectively. After 20 min incubation at 37 °C, the medium was removed and the cells were washed twice in 1 × PBS and harvested. Fluorescent intensity was measured in at least 20,000 cells as suggested by the manufacturer. Data are presented as a geomean ±95% confidence interval. The comparison between the fluorescence intensity in cells with or without doxycycline was done using the two-tailed $t$ test.

**Seahorse assay**. Mitochondrial function and glycolytic flux were determined using Seahorse XFe 96 extracellular flux analyzer (Seahorse, Agilent) as oxygen consumption rate and extracellular acidification rate using "Mito-stress test" or "Glycolysis stress test" methods, respectively. Cells harboring inducible N-Myc overexpression vector were treated ±DOX for 2, 3, 4, or 5 days. Next, cells were plated onto 96-well plate using several strategies to verify experimental outcome. All plating approaches yielded consistently very similar results. In particular, cells were plated at two different concentrations ($4 \times 10^4$ and $2.5 \times 10^4$ cells/well) in multiple wells (i) allowing to attach for 6 h or (ii) overnight to fibronectin-coated wells, or (iii) were attached readily after re-plating using Cell Tak cell adhesive (Corning). FBS-free Seahorse XF RPMI medium (without Phenol Red, Agilent) was supplemented with 2 mM glutamine, 1 mM sodium pyruvate and 2 g/L glucose (for "Mito-stress test") or with 2 mM glutamine (for "Glycolysis stress test"). Measurements were performed at basal state and following serial injections of: oligomycin (1 μM), FCCP (0.5 μM) and rotenone/antymycin A (0.5 μM/1 μM) for "Mito-stress test" or of glucose (10 mM), oligomycin (1 μM), and 2-deoxy-glucose (20 mM) for "Glycolysis stress test".

**TEM analysis of mitochondria**. The samples were fixed with 2.5% glutaraldehyde, 2% paraformaldehyde in 0.1 M sodium cacodylate buffer pH 7.4, and post fixed in 2% osmium tetroxide in 0.1 M cacodylate buffer with 0.3% potassium ferrocyanide for 1.5 h. After rinsing in the same buffer, the tissue was dehydrated through a series of graded ethanol to propylene oxide, infiltrated, and embedded in epoxy resin and polymerized at 70 °C overnight. Semithin sections (0.5 micron) were stained with toluidine blue for light microscope examination. Ultrathin sections (80 nm) were cut and imaged using an FEI Tecnai 200 Kv FEG Electron Microscope with an ATM XR41 Digital Camera. To score mitochondrial damage, 50 images from every sample were selected in which the nucleus and at least five mitochondria were visible at 5 K magnification. Images of these profiles were taken at 29 K with some portion of the nucleus visible, printed and all mitochondria profiles were scored blindly. Mitochondria were scored 0 = normal (aligned cristi and normal outer membrane), 1 = abnormal (swollen and disorganized cristi or disrupted outer membrane) or 2 = very abnormal (both swollen and disorganized cristi and severe outer membrane disruption).

**Gamma H2AX immunostaining**. Cells were harvested and incubated on poly-L-lysin-treated (Sigma, Cat. # 8920) slides for 30 min at 37 °C. The cells were fixed in 4% paraformaldehyde in PBS at 4 °C overnight, blocked by normal donkey serum for 1 h at room temperature and incubated with mouse anti Phospho Ser139-histone H2AX antibody (Millipore, Cat. # 5-636) 1:10,000 overnight at 4 °C. The slides were then washed three times in PBS and incubated with donkey anti-mouse Cy3 IgG antibodies (Millipore, Cat. # AP192C) 1:500 for 1 h at room temperature. The slides were washed three times in PBS, counter-stained with DAPI for 10 min, washed three times in PBS and mounted with Prolong Gold Antifade Mountant (Thermo Fischer, Cat. # P36930).

**Measuring ROS**. ROS was assayed using 2',7'-dichlorofluorescin diacetate (DCFDA) cellular ROS detection assay kit (Abcam # ab113851) or CellRox Green (ThermoFischer # C10444) following manufacturers' protocol. For DCFDA cellular ROS detection assay kit, cells were plated overnight in 96-well plates (25,000 cells per well) in their complete medium with or without doxycycline. The fluorescence excitation/emission 485/535 was measured using the microplate reader

SpectraMax-M5 (Molecular Devices, Sunnyvale, California). For CellRox Green, cells grow in 10-cm plates. CellRox Green was added to the medium 1:500 and incubated for 30 min then cells were harvested and analyzed using flow cytometry.

**Measuring cellular glutathione**. The cells were plated in 96-well plates overnight at a density of 6000 cells per well in their complete medium with or without doxycycline. Glutathione concentration per well was measured using GSH-Glo Glutathione Assay Kit (Promega # V6911) according to the manufacturer's protocol. The luminescence was measured using PHERAstar FSX (BMG Labtech, San Diego, California).

**Drug screening**. Cell screening was done on MYCN-induced and un-induced SKNMM$^{MYCN}$ cells as previously described[54] with the following modifications: to determine optimal cell plating density, SKNMM$^{MYCN}$ cells were plated on a flat-bottomed, white 96-well plate (Corning Cat#3917), at eight different cell densities ranging from 0 cells/well to 10,000 cells/well; 12 wells per cell intensity per plate. Twenty-four hours after plating, medium was changed with or without doxycycline. Medium was changed after that with or without doxycycline every 2 or 4 days. One plate from every testing condition was read every day for 11 days using CellTiter-Glo ('CG', Promega Cat#G7573) on PHERAstar FSX (BMG Labtech, San Diego, California). Cell density of 1250 cells per well and medium change once every 4 days was determined to have good signal to noise separation while maintaining logarithmic growth in both induced and un-induced cells. We tested different dimethyl sulfoxide (DMSO) concentrations for the selected cell density and confirmed minimal cell death with up to 0.197% DMSO. Positive control compound selection was performed with eight positive control compound candidates (doxorubicin HLC, staurosporine, etoposide, SN-38, bortezomib, cyclohex-amide, panobinostat, and TAKA123) arrayed in single-point concentration and 1:3 dilution series. Based on successful cell killing Staurosporine was chosen as positive control for cell screenings. Assay validation in 384-well plates was performed with staurosporine.

**Dose–response curve experiment**. A schematic representation of the dose–response curve experiment is illustrated (Supplementary Data 14). Cells were plated in 96-well plates (Corning Cat#3917) at 1250 cells/100 μL; 12 cell plates per compound plate. Approximately 24 h following plating, medium was changed with fresh medium with or without doxycycline; six-cell plates per a compound plate each. Two days after MYCN induction, medium was changed again with fresh medium with or without doxycycline and cells were drugged with the compound plates and a positive control plates using a Biomek FX (Beckman Coulter) liquid handler equipped with a pin tool. The pin tool transferred 112 nL compound stock, resulting in 890-fold compound dilution. After 3 days of drugging, half of the plates were read. On day 4 post drugging, medium was changed with fresh medium again with or without doxycycline and fresh drugs were added to the medium using the Biomek FX. After 4 days of the second drugging, the other half of the plates were read. The compound plate contained compounds dissolved in DMSO arrayed in a 1:3-fold dilution series in columns 1–10 with eight compounds per plate (target concentration 10 mM or 1 mM). The positive control plate was empty from columns 1–10; in columns 11–12, it contained DMSO and the positive control. Two biological replicates were performed for each 8 days experiment, with three technical replicates per biological replicate.

**Drug screening analysis**. Normalization, fitting, and visualization of dose–response experiments were performed using custom code written in the R programing language (version 3.3.2, R Core Team (2016). R: A language and environment for statistical computing. R Foundation for Statistical Computing, Vienna, Austria. URL https://www.R-project.org/.).

Dose–response curve data analysis for cytotoxic compounds was done as previously described before[54]. Raw CellTiter-Glo (CTG) luminescence signals (RLU) were first log2 transformed, then normalized to the mean of the positive and negative controls on each plate.

In order to detect drugs that reduced MYCN-induced cell death, traditional dose–response analysis was not appropriate because protective drugs would often protect at low concentrations, but eventually induce cytotoxicity at high concentration. To detect a protective effect, we needed to quantify the increase in CTG signal at low concentration before any decrease in signal at high concentration. To do so, the CTG RLUs from all wells were first log2 transformed for each plate, the value for each well was then normalized by subtracting out the mean log2 RLU for negative control wells. The normalized activity of each compound was then fit as a function of the log10 drug concentration using a smooth spline (smooth.spline function in R with default parameters). For each fit, the area under the curve (AUC) for the portion of the spline with RLU above the RLU of the lowest drug concentration tested was calculated. Higher values for this AUC metric indicate greater protection against loss of cell viability as measured by the CTG assay.

**CUX2 expression in SKNMM$^{MYCN}$ cells**. SKNMM$^{MYCN}$ cells were transduced by a lentiviral vector (pLenti-C-mGFP-P2A-Puro) from Origene expressing either CUX2 (Cat# RC222063L4V) or GFP control construct (Cat# PS100093V). GFP-

positive cells were sorted after 48 h and maintained in a selective media containing 1.5-μg puromycin per ml.

**Statistics and reproducibility**. We have made every effort to ensure reproducibility of our data by, when possible, repeating the experiments using independent samples, including positive and negative controls and using multiple approaches to confirm our observations. We stated the sample number for each experiment and how many times each experiment was repeated in the figure legends. However, the following experiments were done once:

Fig. 1f–i: This was a unique case in which there was a discrepancy between COG FISH report and next-generation sequencing report. This sample was carefully studied as described in the results section.

Fig. 3b: This is the first and only O-PDX developed from *ATRX*-mutant neuroblastoma, as explained in the text.

Fig. 5h: This experiment was done to confirm the high-throughput screening and was done in a dose response.

Fig. 5i: This experiment was done once. However, the ability of RA to reduce MYCN level is well-established in literature[70].

Supplementary Fig. 1B–S1E: Was done to all the clinical samples as mentioned in the material and methods.

Supplementary Fig. 4H. This experiment was done once. However, this experiment is another evidence for the increase in replicative stress when ATRX is inactivated and MYCN is ectopically expressed in different cells as shown in Fig. 5.

Supplementary Fig. 7C: This experiment was done once as another readout for increase ROS shown in Supplementary Fig. 7D.

**Reporting summary**. Further information on research design is available in the Nature Research Reporting Summary linked to this article.

## Data availability

Whole-genome bisufite sequencing data and ChIP-seq data have been deposited in the EMBL-EBI database under the accession code EGAS00001003257. Whole-exome sequencing data have been deposited in the EMBL-EBI database under the accession code EGAS00001002528. All of our extensive epigenetic data and analysis are freely available in a cloud-based viewer (https://pecan.stjude.cloud/proteinpaint/study/mycn_nbl_2018). All O-PDX tumors described here are freely available with no obligation to collaborate through the Childhood Solid Tumor Network (http://www.stjude.org/CSTN/). We downloaded G4 Motifs from supplementary data of Du et al. 2009[59]. We downloaded 169,222 R-Loop domains in genic and proximal regions from (http://rloop.bii.a-star.edu.sg/). All the other data supporting the findings of this study are available within the article and its Supplementary Information files and from the corresponding author upon reasonable request.

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

## Acknowledgements

We thank Angie McArthur for editing the manuscript and Kevin Freeman for the MYCN-inducible constructs. The NBL-S cell line was provided by Garrett Brodeur. This work was supported, in part, by Cancer Center Support (CA21765) from the NCI, grants to M.A.D. from the NIH (EY014867, EY018599, and CA168875) and ALSAC. M.A.D. was also supported by a grant from Alex's Lemonade Stand Foundation for Childhood Cancer, the Tully Family Foundation, and the Peterson Foundation. E.S. was supported by the St. Baldrick's Foundation. The statistical work was supported by NIH/NCI grant U10 CA180899 (Children's Oncology Group Statistics and Data Center). Most of this research was supported by the Howard Hughes Medical Institute. WGS, WGBS, and RNA-seq were performed as part of the St. Jude Children's Research Hospital—

Washington University PCGP. Preclinical studies were performed with assistance from the Animal Imaging Shared Resource; electron microscopy was performed with assistance from the Cell and Tissue Imaging Shared Resource; histopathologic analysis was performed with assistance from the Veterinary Pathology Shared Resource (all of St. Jude).

## Author contributions

M.Z., S.F., X.C., and M.A.D. conceived and designed the experiments and analysis. M.Z., E.S., J.J., L.G., R.N., J.N., J.E., H.M., D.Y., Y.L., M.M.K., M.V., S.M.P.-M., M.R.C., A.B., P.V., S.L., A.S., J.F.S., and M.P.M. collected the data. M.Z., Y.F., B.X., X.C., S.F., J.W., C.V.R., A.N., and M.A.D. performed analysis. M.D.H., R.E.G., and S.M.P.-M. provided samples and reagents. S.F., A.P., J.Z., E.R.M., R.K.W., S.H., J.R.D., and M.A.D. provided project leadership and guidance. X.Z. developed the visualization tools and web portal. M.Z., S.F., X.C., A.P., and M.A.D. wrote the paper.

## Competing interests

The authors declare no competing interests.
