## [Peer Review File · Nature Communications]

Reviewers' comments:

Reviewer #1 (Remarks to the Author):

The paper by Zeineldin and co-workers describes the results of investigation of the mutual exclusive occurrence of MYCN amplification and ATRX defects in neuroblastoma.

Overall, this is an impressive amount of work conducted by the teams. First, the data our summarized on 473 patients with focus on ATRX genomic defects (mainly intragenic deletions) which are further confirming previous findings in relation to clinical features and genomic context for ATRX defective tumors but here further supported by the large numbers of patients and extensive statistics. Next, the presumed incompatibility of MYCNamp and ATRX deficiency was investigated in a mouse model for neuroblastoma showing complete loss of tumor formation in MYCN only mice whereas there is strong suppression of tumor suppression (but not complete) in MYCN/ALKmut mice (see comments). To investigate the underlying nature of this observation, several experiments were executed. First, in 12 cancer cell lines including also neuroblastoma cell lines with or without MYCNamp were evaluated after ATRX knock down with no effects on telomere status, cell cycle and growth except for MYCNamp lines showing decreased colony formation. Next, CRISP ko wadsone using 2 gRNAs showing loss of the mutant alleles in MYCNamp lines after 10 days. To test effects of reconstitution, MYCN was inducibly overexpressed in ATRX deficient lines and WT lines (with and without MYCNamp with ATRX deficient lines showing dramatic loss of cells after 6 days of MYCN overexpression which was also tested in an in vivo competition model. Next, an impressive genomic analysis was done to identify underlying molecular targets by generating multiple layers of genomics data (methylation, expression, chromatin) in 8 cell lines and SKNMMmycn cells with a viewer to check these data. Using this approach pathways involved in metabolism and mitochondrial gene expression were found to be enriched. Further analysis using metabolomics and analysis of mitochondrial function supported this finding. Increased ROS levels were shown as well as evidence for genomic instability and increased sensitivity for hydroxy urea, a known inducer of replicative stress. Finally, drug screening identified multiple drug implicated in DNA damage, DNA repair and check points sensitizing for MYCN upregulation in ATRX deficient cells.

The authors conclude that metabolic reprogramming and mitochondrial dysfunction contribute to ROS and subsequent replicative stress.

Comments:

1. While I agree that the presented data support the possibility that ATRX loss in combination with MYCN overexpression could lead to synthetic lethality through the described processes, I lack some further experiments exploring the role of ATRX in ensuring smooth replication at G4 sequences. First, two possible ways to demonstrate replicative stress are staining or protein blotting for pRPA32 and DNA combing (Zeman, M. K., & Cimprich, K. A. (2013). *Nature Cell Biology*, 16(1), 2–9), or alternatively EDU-seq (see Macheret, M., & Halazonetis, T. D. (2018). *Nature Protocols*, 14(1), 51–67). To further test a possible impact of MYCN overexpression on G4 related induced replicative stress, further tests with G4 ligands (eg TYMPyP4) and G4 antibodies could be used. A further possiblity which cannot be excluded to contribute to the observed lethality are replication transcription conflicts which can be observed through increased R-loop formation. In fact, the authors make a statement in relation to the possible role of G4 DNA but given the well-known function of ATRX in this context, some key experiments to explore this should be added.
2. I would suggest to comment in more detail on the observed results in the mouse models. While the conclusions are valid, it is interesting to see that in the more aggressive MYCN/ALKmut model, ATRX loss does not completely exclude tumor formation, suggesting that additional ALK signaling partially rescues the proposed synthetic lethality.
3. Figure 5D spectral karyotyping. What is the reference karyotype, some more annotation here is

needed to clarify the findings?

Reviewer #2 (Remarks to the Author):

This is an interesting article by Zeineldin et al, which examines the underlying basis for the distinct mutually exclusivity of MYCN and ATRX alterations in Neuroblastoma. Prior studies have shown that ATRX mutations and MYCN are mutually exclusive events with ATRX largely seen in older children with much poorer disease trajectory. In this study, the authors validate these observations in both stage 3 and 4 NB using detailed analyses of MYCN copy number and ATRX mutations in a large collection of NB from prospective trials. The authors then exploited conditional inactivation of ATRX in murine models and human NB cell lines to examine the functional significance of MYCN and ATRX loss in NB cells.

The authors demonstrate using genetic models that loss of ATRX intriguingly confers a survival advantage in 2 NB mouse models, and use in vivo "mixing experiments" to demonstrate high MYCN levels are incompatible with ATRX mutations. They used loss and gain of function experiments in a panel of high and low MYCN NB cells and showed ATRX loss diminishes cell viability and transforming phenotypes in MYCN amplified NB cell lines.

In addition, overexpression of MYCN in ATRX mutant cell lines had the same effect.

They showed through a series of experiment using an ATRX mutant NB cell line with inducible MYCN (SKNMM-MYCN) that MYCN associated metabolic/glutamine addiction correlates with altered mitochondrial structure, increased ROS, replicative stress and DNA damage. They confirmed using a drug screen that SKNMM -MYCN cells were particularly sensitized to DNA damage/replicative pathway agents, as well as retinoids. Based on these observations they propose increased ROS/replicative stress induced by MYCN may enhance replicative stress of ATRX mutant cells leading to MYN/ATRX synthetic lethality.

Major comments:

The finding of synthetic lethality between MYCN and ATRX mutations is a novel and intriguing finding that is generally well supported by the evidence provided by the authors. The authors provide substantial data aimed at delineating mechanisms, which suggest MYCN/ATRXmut synthetic lethality may reflect MYCN- mediated augmentation of replicative stress in ATRX mutant cells. Although the authors provide voluminous data that is generally supportive of their hypothesis, some of the data presented requires more clarification.

Specific comments:

1. Figure 1 show that while a majority of NB has CNAs leading to ATRX gene deletions – there are a number of patients in which ATRX mutations are predicted to spare much of the coding protein. Did the authors examine whether degrees of ATRX protein loss (complete or predicted partial) have any correlation with disease phenotypes?

2. The in vivo experiments to support the contention that ATRX mutations and high MYCN are incompatible are not well detailed in the manuscript. The construction and validation of the ATRX mutant/MYCN murine is lacking in the manuscript (Figure 1K) – I assume the ATRX gene was completely deleted? Similarly in Fig 2 – important details are missing in the mixing experiments – did the authors re-isolate the cells to confirm the status of ATRX and MYCN?

3. On page 7, the authors mention that ALT did not accompany ATRX k/d across all cell lines – can they authors elaborate on how this may or may not be related/relevant to their primary observation of replicative stress in ATRX mutant cells?

4. In figure 2 c, it appears that the ATRX k/d in SKNMBE2 was not successful – this should be

commented on or removed. In the colony formation assays only data for NB line –NB-5 is shown. The bar graph does not seem to quantitatively correlate with the plate image –are the numbers in plot C – sum of multiples?

5. In figure 4, the authors implicate mitochondrial damage and oxidative stress in MYCN induced SKNMM cells. Is this a cell line specific phenomenon? Have the authors compared the mitochondrial architecture in ATRT wild type cells, or MYCN amp/non ATRX mutant cells?

6. In Figure 5, the authors demonstrate evidence of DDR in SKNMMmycn cells; again – is this cell lines restricted. As the authors propose a baseline replicative stress in ATRX mutant cells, it would have been informative to quantify the extent of DDR across the various NB cell lines with and without endogenous or engineered ATRX mutations and MYCN amp/increased expression. This will help strengthen their conclusion of synthetic lethality.

7. The authors performed extensive epigenomic analyses and 2 drug screens to map down stream effectors of MYCN/ATRX synergy. While the identification of retinoids and CUX2 is interesting, the significance and implications for understanding MYCN/ATRX synergy from these studies is not clearly articulated.

Overall, I found this to be an interesting manuscript with an innovative finding. The authors include substantial amounts of detailed data - generally supportive of their premise, however, the rationale for execution and inclusion of specific experiments especially the drug screen was not clear. The authors may want to exploit their substantial data to better clarify more direct mechanisms to support their observation

Reviewer #3 (Remarks to the Author):

This study “MYCN amplification and ATRX mutations are incompatible in neuroblastoma” by Zeineldin et al reports the mutually exclusive relationship between amplification of the oncogene MYCN and mutation of the tumor suppressor ATRX in human neuroblastoma, and investigates the plausible molecular mechanism leading to it. They find that MYCN amplification causes metabolic reprogramming including enhanced Warburg effect, increased uptake of both glucose and glutamine, dysfunctional mitochondria and increased oxidative stress in the ATRX mutant cells. In general, the conclusions reached by the authors from the metabolic profiling experiments are supported by the current data. However, the presentation of the results could be improved. Specific comments are listed below.

1. Since M+4 is the most abundant mass isotopomer of aspartate, it would be helpful to also include it in the diagram in Figure 4A.

2. It is not clear to me why only changes in M+5 glutamate and M+5 alpha-ketoglutarate are emphasized by the arrows in Figure 4B. Is that because all other changes are not statistically significant? If so, this needs to be clarified in the figure legend.

Minor:

Page 13: ‘Cells also upregulate pathways required to mitigate the reactive oxygen species (ROS) that are a natural biproduct of mitochondrial metabolism to prevent excessive protein or DNA damage’: ‘biproduct’ should be ‘byproduct’.

Point-by-point response to reviewers:
REVIEWER #1

The paper by Zeineldin and co-workers describes the results of investigation of the mutual exclusive occurrence of MYCN amplification and ATRX defects in neuroblastoma.

Overall, this is an impressive amount of work conducted by the teams. First, the data our summarized on 473 patients with focus on ATRX genomic defects (mainly intragenic deletions) which are further confirming previous findings in relation to clinical features and genomic context for ATRX defective tumors but here further supported by the large numbers of patients and extensive statistics. Next, the presumed incompatibility of MYCNamp and ATRX deficiency was investigated in a mouse model for neuroblastoma showing complete loss of tumor formation in MYCN only mice whereas there is strong suppression of tumor suppression (but not complete) in MYCN/ALKmut mice (see comments). To investigate the underlying nature of this observation, several experiments were executed. First, in 12 cancer cell lines including also neuroblastoma cell lines with or without MYCNamp were evaluated after ATRX knock down with no effects on telomere status, cell cycle and growth except for MYCNamp lines showing decreased colony formation. Next, CRISPR ko was done using 2 gRNAs showing loss of the mutant alleles in MYCNamp lines after 10 days. To test effects of reconstitution, MYCN was inducibly overexpressed in ATRX deficient lines and WT lines (with and without MYCNamp with ATRX deficient lines showing dramatic loss of cells after 6 days of MYCN overexpression which was also tested in an in vivo competition model. Next, an impressive genomic analysis was done to identify underlying molecular targets by generating multiple layers of genomics data (methylation, expression, chromatin) in 8 cell lines and SKNMMmycn cells with a viewer to check these data. Using this approach pathways involved in metabolism and mitochondrial gene expression were found to be enriched. Further analysis using metabolomics and analysis of mitochondrial function supported this finding. Increased ROS levels were shown as well as evidence for genomic instability and increased sensitivity for hydroxy urea, a known inducer of replicative stress. Finally, drug screening identified multiple drug implicated in DNA damage, DNA repair and check points sensitizing for MYCN upregulation in ATRX deficient cells. The authors conclude that metabolic reprogramming and mitochondrial dysfunction contribute to ROS and subsequent replicative stress.

Comments:

1. While I agree that the presented data support the possibility that ATRX loss in combination with MYCN overexpression could lead to synthetic lethality through the described processes, I lack some further experiments exploring the role of ATRX in ensuring smooth replication at G4 sequences. First, two possible ways to demonstrate replicative stress are staining or protein blotting for pRPA32 and DNA combing (Zeman, M. K., & Cimprich, K. A. (2013). *Nature Cell Biology*, 16(1), 2–9), or alternatively EDU-seq (see Macheret, M., & Halazonetis, T. D. (2018). *Nature Protocols*, 14(1), 51–67). To further test a possible impact of MYCN overexpression on G4 related induced replicative stress, further tests with G4 ligands (eg TYMPyP4) and G4 antibodies could be used. A further possibility which cannot be excluded to contribute to the observed lethality are replication transcription conflicts which can be observed through increased R-loop formation. In fact, the authors make a statement in relation to the possible role of G4 DNA but given the well-known function of ATRX in this context, some key experiments to explore this should be added.

We thank reviewer #1 for his/her comments and suggestions. Previous publications have shown that ATRX plays a role in reducing DNA replicative stress¹⁻⁴. Here, we extend those studies to the synthetic lethality phenotype observed in neuroblastoma. We found that γ H2AX were increased by immunoblotting and immunofluorescence, DNA fragmentation was increased by spectral karyotyping and COMET and

cells were more sensitive to hydroxyurea and other pharmacological agents that induce DNA replicative stress. To extend those findings, we performed additional experiments on 5 other ATRX-mutant cell lines including 2 glioma cell lines that had isogenic ATRX-wild type controls (Fig. S4). We appreciate the suggestion to include immunoblotting for pRPA32 and those data have been added to Fig. S4. An example of the isogenic lines (U251 and MOG-G-UVW) is presented below showing clear upregulation of pRPA32 in this set of isogenic lines.

ATRX inactivation and MYCN overexpression induce replicative stress. The isogenic glioma cell lines were engineered to have Dox-inducible MYCN transgene. Knocking-out ATRX and ectopic MYCN expression increases the levels of phospho-RPA32.

We also extended our results beyond MYCN to MYC. None of the ATRX mutant cell lines have a canonical MYC signature so we have made stable cell lines in SKNMM, GM847, SAOS2, U2OS and WI-38 cells that are all ATRX mutant that have inducible expression of MYC. The experiments were identical to those for induction of MYCN and the results were identical. Therefore, elevated MYC or MYCN can lead to synthetic lethality in ATRX mutant cells.

The text now reads: “ These data were extended to 5 other ATRX-mutant cell lines including 2 glioma cell lines that had isogenic ATRX-wild type controls (Fig. S4)²⁷. A marker of DNA replicative stress (pRPA32) was upregulated in the absence of ATRX and in the presence of doxycycline (Fig. S4). A similar experiment performed with MYC gave the same results (data not shown).”

Regarding the role of MYCN in G4 structures and the role of R loops, we directly tested the deposition of histone H3.3 at G4 structures using ChIP-seq and compared those data to our extensive epigenetic profiling including MYCN ChIP-seq. We used computational methods to identify R loops that overlap with the H3.3 and G4 structures. We have added those data to Fig. 6, the cloud-based portal (https://pecan.stjude.cloud/proteinpaint/study/mycn_nbl_2018) and the results now read:

“Heterochromatic H3.3 deposition is altered in ATRX-mutant NB
ATRX plays an important role in H3.3 deposition in chromatin⁴⁰. In addition to telomeres and pericentric heterochromatin, ATRX helps to resolve G-quadruplex (G4) structures in the DNA by deposition of H3.3 at those sites across the genome^{8,40,41}. This is important because G4 structures can inhibit DNA replication and transcription^{40,42,43}. G4 structures often overlap with noncanonical DNA:RNA hybrids called R-loops because both structures are favored in G rich regions of the genome under negative torsional tension (e.g. active promoters)⁴⁴⁻⁴⁶. In cancer cells, R loops can contribute to DNA breaks and genome instability⁴⁷⁻⁴⁹. To determine if the pattern of H3.3 deposition is altered in ATRX-mutant NB, we performed Cut & Run⁵⁰⁻⁵² on 6 NB cell lines divided into 3 groups: MYCN-amplified cells (SKNBE2 and IMR32), ATRX-mutant cells (SKNMM and CHLA90) and MYCN non-amplified/ATRX wild type cells (LAN6 and SKNFI). We identified the reproducible peaks in each group (ATRX, MYCN and WT) and created a reference list of 51,919 H3.3 peaks across our dataset. Among those peaks, 18,436 (35%) overlapped with DNA sequences that are predicted to form G4 structures and 70% (12,932) of those are predicted to form R loops. As expected, there were fewer H3.3 peaks in the ATRX group and nearly a 2-fold reduction in H3.3 peaks overlapping G4 sequences (Fig. 6A). For every H3.3 peak in the dataset, we assigned it to a group defined by the pattern across the samples (ATRX, MYCN and WT). The constitutive (C) group includes H3.3 peaks found at the same location across all samples (ATRX, MYCN and WT) (Fig. 6B). The enriched (E) group includes H3.3 peaks that are higher in ATRX than MYCN and WT (Fig. 6B). The depleted (D) group includes H3.3 peaks that are lower in ATRX than in MYCN and WT and the overlapping (O) group has peaks in ATRX and either MYCN or WT (Fig. 6B). The D group was the largest (Fig. 6B) and 2,428 of those had a correlation between ChromHMM state and the presence or absence of H3.3 peaks; 514 (21%) had G4 sequences and 69% of those (357) were predicted to form R loops. The majority of H3.3 peaks that were depleted in ATRX mutant NB relative to MYCN and WT that had a correlation with ChromHMM state were in enhancers while those with G4 sequences (and R loops) were in promoters (Fig. 6C). For both the non-genic and genic regions, there was a shift from active euchromatic states (states 1-4) to inactive heterochromatic states (states 15-17) in the ATRX group relative to MYCN and WT (Fig. 6D-H). There was overlap between H3.3 peaks and CTCF sites (state 18) consistent with the role of H3.3 in CTCF binding to establish insulated chromatin domains⁵³⁻⁵⁵ (Fig. 6D-H). We performed pathway analysis on the genes that had promoters or enhancers that were depleted for H3.3 in ATRX mutant NB and had correlation with ChromHMM states and reduced expression in ATRX mutant relative to MYCN and WT. The most significant pathways were involved in neuronal differentiation and neural development (Table S11). For example, the DUSP26 gene is a MYCN target that has G4 sequences in the promoter and selective deposition of H3.3 in MYCN amplified NB relative to ATRX-deficient NB cells (Fig. 6G). DUSP26 is a phosphatase that is expressed in neuroendocrine cells and can inhibit neuronal differentiation^{56,57}. It has also been shown to dephosphorylate and inhibit p53 in neuroblastoma cells⁵⁸. We performed the same H3.3 ChIP-seq on SKNMM^{MYCN} with and without doxycycline and found similar chromatin changes at MYCN target genes (Fig. 6H). Taken together, our data show that there is dramatic reorganization of the chromatin landscape in ATRX mutant NB beyond telomeres and centromeres. This is particularly notable at G4 sequences in promoters and enhancers and may impact the ability of ATRX mutant NB cells to differentiate relative to those with wild type ATRX.”

We agree with the reviewer that the inability of the ATRX-mutant cells to resolve R-loops in the genome may contribute to the synthetic lethality phenotype in MYCN-amplified cells. It was recently shown that ATRX reduces R-loop formation by depositing H3.3 at the G4 structures in the repetitive regions of the genome. The H3.3 ChIP-seq data described above are consistent with this model.

The discussion now reads:

“ While the inability of ATRX mutant neuroblastoma cells to resolve G4 structures promotes tumorigenesis by preventing differentiation, it also causes DNA replicative stress and slows tumor growth. Indeed, it was recently shown that ATRX can play a role in suppressing R-loop (RNA-DNA hybrid) formation¹⁹. R-loops can lead to collapse of the DNA replication fork and replicative stress^{20,21}. This may be why ATRX mutant neuroblastomas are slow growing and indolent. DNA replicative stress sensitizes ATRX mutant neuroblastomas to drugs that further exacerbate DNA repair such as ATR inhibitors²². It also leads to synthetic lethality with ectopic MYCN expression because of the ROS and DNA replicative stress that results from metabolic reprogramming induced by MYCN. This genetic synthetic lethality may eventually be exploited in the clinic by reducing the function of ATRX in patients with MYCN-amplified tumors. Alternatively, inducing metabolic (or other changes) caused by MYCN expression in ATRX-mutant tumors could be useful. To achieve this ambitious goal of generating synthetic lethality in neuroblastoma cells in patients with high-risk disease by targeting these 2 pathways, much will need to be learned about the downstream targets of ATRX and MYCN that contribute to this phenotype. “

2. I would suggest to comment in more detail on the observed results in the mouse models. While the conclusions are valid, it is interesting to see that in the more aggressive MYCN/ALKmut model, ATRX loss does not completely exclude tumor formation, suggesting that additional ALK signaling partially rescues the proposed synthetic lethality.

We analyzed the expression of wild type (unfloxed) and inactivated (floxed) alleles of *Atrx* in tumors and normal tissue (liver) from mice in the study. Representative data for 5 mice are shown below indicating that those with tumors had evidence of wild type *Atrx*. This suggests that they escaped *Atrx* inactivation and synthetic lethality. We were also concerned about the loss of the cell of origin for neuroblastoma in these genetic models so we analyzed the adrenals from those mice. The results section has now been modified to read:

“ To determine if ATRX mutations and MYCN amplification are incompatible *in vivo*, we conditionally inactivated ATRX in 2 genetically engineered mouse models of neuroblastoma (Fig. 1I-L)^{2,23,24}. One model (LSL-MYCN;Dbh-iCre) is a transgenic line with conditional expression of MYCN from the *Rosa26a* locus mediated by Cre expressed in the dopaminergic cells of the sympathoadrenal lineage²². The other model includes the Th-ALK^{F1174L} transgene that potentiates MYCN-mediated tumorigenesis²³. For each of these models, tumor formation was reduced and survival was significantly increased ($p < 0.0001$) when *Atrx* was simultaneously inactivated with elevated MYCN expression (Fig. 1K-L). There was a significant increase in LSL-MYCN;Dbh-iCre female mice with tumors ($p=0.0017$) but none of the *Atrx*^{Lox/Lox};LSL-MYCN;Dbh-iCre mice (male or female) developed tumors. For the mice with the ALK^{F1174L} transgene, there was no difference in male/female ratio. The few tumors that did form in the *Atrx*^{Lox/Lox};LSL-MYCN;Dbh-iCre;ALK^{F1174L} mice all had evidence of an intact *Atrx* allele suggesting that rare cells that did not undergo *Atrx* inactivation contributed to tumor formation (data not shown). It is also possible that ALK signaling partially rescued the synthetic lethality. *Atrx* inactivation has been shown to reduce cell survival in brain, muscles and testes^{2,25,26}. Histopathological review of adrenal gland and paravertebral sympathetic ganglia from LSL-MYCN;Dbh-iCre and *Atrx*^{Lox/Lox};LSL-MYCN;Dbh-iCre mice at 3 weeks and 1 year of age showed no differences (Fig. S2) suggesting that the lack of tumor formation was not due to death of the NB cell of origin.”

Detection of floxed Atrx in tumors. RT-PCR was performed on cDNA from RNA extracted from the tumor and liver of individual mice. The floxed allele is 307 bp and the unfloxed allele is 417 bp. Each of the tumors in these mice had evidence of unfloxed Atrx alleles suggesting that they escaped Atrx inactivation.

3. Figure 5D spectral karyotyping. What is the reference karyotype, some more annotation here is needed to clarify the findings?

We apologize for the omission. We have now included a representative reference SKY for the parental line (–DOX) to compare to the line with DOX to Fig. 5D. Arrows indicate new chromosomal lesions and there are a large number of derivative chromosomal fragments labeled “m” on the SKY in the lower panel that had DOX. The conclusion from our cytogenetics laboratory in their report was that the sample with DOX had a massive increase in chromosomal breaks.

Reviewer #2 (Remarks to the Author):

This is an interesting article by Zeineldin et al, which examines the underlying basis for the distinct mutually exclusivity of MYCN and ATRX alterations in Neuroblastoma. Prior studies have shown that ATRX mutations and MYCN are mutually exclusive events with ATRX largely seen in older children with much poorer disease trajectory. In this study, the authors validate these observations in both stage 3 and 4 NB using detailed analyses of MYCN copy number and ATRX mutations in a large collection of NB from prospective trials. The authors then exploited conditional inactivation of ATRX in murine models and human NB cell lines to examine the functional significance of MYCN and ATRX loss in NB cells.

The authors demonstrate using genetic models that loss of ATRX intriguingly confers a survival advantage in 2 NB mouse models, and use in vivo “mixing experiments” to demonstrate high MYCN levels are incompatible with ATRX mutations. They used loss and gain of function experiments in a panel of high and low MYCN NB cells and showed ATRX loss diminishes cell viability and transforming phenotypes in MYCN amplified NB cell lines.

In addition, overexpression of MYCN in ATRX mutant cell lines had the same effect.

They showed through a series of experiment using an ATRX mutant NB cell line with inducible MYCN (SKNMM-MYCN) that MYCN associated metabolic/glutamine addiction correlates with altered mitochondrial structure, increased ROS, replicative stress and DNA damage. They confirmed using a drug screen that SKNMM –MYCN cells were particularly sensitized to DNA damage/replicative pathway agents, as well as retinoids. Based on these observations they propose increased ROS/replicative stress induced by MYCN may enhance replicative stress of ATRX mutant cells leading to MYN/ATRX synthetic lethality.

Major comments:

The finding of synthetic lethality between MYCN and ATRX mutations is a novel and intriguing finding that is generally well supported by the evidence provided by the authors. The authors provide substantial data aimed at delineating mechanisms, which suggest MYCN/ATRX mut synthetic lethality may reflect

MYCN- mediated augmentation of replicative stress in ATRX mutant cells. Although the authors provide voluminous data that is generally supportive of their hypothesis, some of the data presented requires more clarification.

Specific comments:

1. Figure 1 show that while a majority of NB has CNAs leading to ATRX gene deletions – there are a number of patients in which ATRX mutations are predicted to spare much of the coding protein. Did the authors examine whether degrees of ATRX protein loss (complete or predicted partial) have any correlation with disease phenotypes?

We thank the reviewer for this comment and we agree that it is an interesting question. In the original study published in JAMA in 2012 (<https://www.ncbi.nlm.nih.gov/pubmed/22416102>), we had access to detailed clinical records for all the patients in the study. In that cohort, there was no difference in the relationship between the predicted (in frame deletion versus missense or nonsense) or actual (IHC) degree of ATRX protein loss and disease phenotype (indolent course of disease). Unfortunately, we do not have access to detailed clinical records for the COG cohort in this manuscript. We only have outcome data that is presented in Fig. 1. The indolent course of disease that is characteristic of ATRX mutant NB would require more in-depth analysis of a patient population (adolescents and young adults) that is often underserved and difficult to track through cooperative group studies such as those done by COG. We have modified the discussion as follows:

“ In the original study describing ATRX mutations in neuroblastoma^{27,28}, there was no difference in outcome or clinical presentation for patients with in-frame deletions versus missense or nonsense mutations. However, that cohort was relatively small and a much larger study of ATRX-mutant neuroblastomas would be required to determine if there is any genotype-phenotype correlation for the type of ATRX mutation.”

2. The in vivo experiments to support the contention that ATRX mutations and high MYCN are incompatible are not well detailed in the manuscript. The construction and validation of the ATRX mutant/MYCN murine is lacking in the manuscript (Figure 1K) – I assume the ATRX gene was completely deleted? Similarly in Fig 2 – important details are missing in the mixing experiments – did the authors re-isolate the cells to confirm the status of ATRX and MYCN?

We apologize for this omission. We used the LSL-MYCN;Dbh-iCre mouse model that was developed in the Schulte lab^{23,29}. Briefly, Cre is expressed in the sympathoadrenal lineage from the dopamine beta hydroxylase (Dbh) promoter. The Lox-Stop-Lox (LSL) cassette is excised by Cre at the Rosa26 locus and MYCN is expressed. The advantage of this mouse model is that it has higher penetrance of neuroblastoma across genetic backgrounds and it recapitulates the molecular and cellular features of human neuroblastoma. We validated efficient Cre excision of the LSL by PCR (see below).

Detection of floxed LSL-MYCN in tumors. PCR of genomic DNA extracted from tumors and liver (control) with primers that are specific for the Cre-recombined allele of LSL-MYCN in the Rosa26 locus. The faint band in the liver samples may be due to low level leaky expression of Cre.

The $Atrx^{Lox}$ mouse strain was developed in the Pickett lab and has also been described previously^{2,25}. Cre expression leads to deletion of *Atrx* exon 18 resulting in a frameshift and loss of *Atrx* protein. We have validated the recombination of $Atrx^{Lox}$ allele by PCR as shown below:

Detection of floxed $Atrx^{Lox/Lox}$ allele in tumors. PCR of genomic DNA extracted from the supraadrenal gland of $Atrx^{Lox/Lox}$ mice with and without Cre.

The results read:

“ To determine if ATRX mutations and MYCN amplification are incompatible in vivo, we conditionally inactivated ATRX in 2 genetically engineered mouse models of neuroblastoma (Fig. 1I-L)^{11,22,23}. One model (LSL-MYCN;Dbh-iCre) is a transgenic line with conditional expression of MYCN from the Rosa26a locus mediated by Cre expressed in the dopaminergic cells of the sympathoadrenal lineage²². The other model includes the Th-ALK^{F1174L} transgene that potentiates MYCN-mediated tumorigenesis²³. For each of these models, tumor formation was reduced and survival was significantly increased ($p < 0.0001$) when *Atrx* was simultaneously inactivated with elevated MYCN expression (Fig. 1K-L).”

For the mixing experiment in Figure 2, one group (Fig. 2N,O) of mice were injected with a mixture of unlabeled SKNMM^{MYCN} and labeled (YFP and luciferase) SKNMM^{CONT} cells. The other group (Fig. 2P,Q) was labeled SKNMM^{MYCN} and unlabeled SKNMM^{CONT} cells. We monitored tumor growth with xenogen imaging (luciferase) and ultrasound. The tumors that developed in the mice from the first group (Fig. 2N,O) were luciferase and YFP positive while the tumors that developed in the second group (Fig. 2P,Q) were luciferase and YFP negative. YFP was analyzed by flow cytometry. These data suggest that the SKNMM^{CONT} cells can form a tumor but not the SKNMM^{MYCN} cells. To confirm those data, we also performed molecular analyses on these tumors as shown in the figure below:

Molecular validation of growth competition experiment in vivo. PCR of the tumors that developed in the mixing experiment for groups 1 and 2 described above and in Fig. 2N-Q. The upper gel shows results of PCR using primers specific for the MYCN transgene and the lower gel is primers specific for the CONT vector. In both groups, only the CONT cells grew. Positive control cell lines are on the right.

3. On page 7, the authors mention that ALT did not accompany ATRX k/d across all cell lines – can they authors elaborate on how this may or may not be related/relevant to their primary observation of replicative stress in ATRX mutant cells?

The experiments described on page 7 were short term experiments and we did not expect ALT after just a few rounds of cell division. We mention it in the results section because our manuscript is focused on a dual role of ATRX inactivation in neuroblastoma tumorigenesis. To clarify this point, the discussion now reads:

“ Based on data presented here, we propose that ATRX mutation contributes to tumorigenesis in two ways. First, defects in H3.3 deposition at telomeres leads to telomere maintenance through ALT. Second, ATRX mutations lead to defects in H3.3 deposition at G4 structures in promoters and enhancers of genes involved in neuronal differentiation including retinoic acid responsive genes. As a result, gene expression is attenuated and ATRX mutant neuroblastoma cells continue to proliferate. While DAXX mutations can also contribute to ALT, they do not alter H3.3 deposition at G4 structures. We propose that both mechanisms (ALT and G4 resolution) are essential for tumorigenesis and that is why we did not detect any DAXX mutations in our cohort.

While the inability of ATRX mutant neuroblastoma cells to resolve G4 structures promotes tumorigenesis by preventing differentiation, it also causes DNA replicative stress and slows tumor growth. This may be why ATRX mutant neuroblastomas are slow growing and indolent. DNA replicative stress sensitizes ATRX mutant neuroblastomas to drugs that further exacerbate DNA repair such as ATR inhibitors⁵⁷. It also leads to synthetic lethality with ectopic MYCN expression because of the ROS and DNA replicative stress that results from metabolic reprogramming induced by MYCN. This genetic synthetic lethality may eventually be exploited in the clinic by reducing the function of ATRX in patients with MYCN-amplified tumors. Alternatively, inducing metabolic (or other changes) caused by MYCN expression in ATRX-mutant tumors could be useful. To achieve this ambitious goal of generating synthetic lethality in neuroblastoma cells in patients with high-risk disease by targeting these 2 pathways, much will need to be learned about the downstream targets of ATRX and MYCN that contribute to this phenotype. “

In addition, we have added new cell lines to the revised manuscript. Those lines (U251 and MOG) have isogenic pairs of ATRX^{+/+} and ATRX^{-/-}. In both lines, ectopic expression of MYCN leads to synthetic lethality in the ATRX^{-/-} but not the ATRX^{+/+}. More importantly, the U251 line that is ATRX^{-/-} has ALT but the MOG ATRX^{-/-} line does not. This shows that synthetic lethality is not dependent on ALT and lends additional support for your hypothesis that ATRX inactivation does more than just help cancer cells to maintain telomeres through ALT.

4. In figure 2 c, it appears that the ATRX k/d in SKNMBE2 was not successful – this should be commented on or removed. In the colony formation assays only data for NB line –NB-5 is shown. The bar graph does not seem to quantitatively correlate with the plate image –are the numbers in plot C – sum of multiples?

We thank the reviewer for this important point. Knockdown of ATRX with shRNA was incomplete and the SKNBE2 cells were sensitive to CRISPR inactivation of ATRX. Therefore, we believe it was the suboptimal reduction in ATRX from the shRNA experiments that led to the difference in phenotype. As suggested, we removed the SKNBE2 data from Fig. 2C. The data in the bar graph for NB-5 are mean and standard deviation from multiple biological replicates.

5. In figure 4, the authors implicate mitochondrial damage and oxidative stress in MYCN induced SKNMM cells. Is this a cell line specific phenomenon? Have the authors compared the mitochondrial architecture in ATRX wild type cells, or MYCN amp/non ATRX mutant cells?

We extended our EM analysis to U2OS^{MYCN} cells to have another example of an ATRX mutant line and SKNBE2^{MYCN} which are MYCN amplified and ATRX wild type. In Fig. 2I, The U2OS cells show synthetic lethality and SKNBE2 are unaffected. As for the SKNMM^{MYCN} cells, in the presence of DOX, the U2OS cells had a dramatic decrease in normal mitochondria and increase in disrupted mitochondria using the same blinded scoring approach as for SKNMM^{MYCN} (Fig. 4L-O). The SKNBE2 were unaffected. Those data have been added to Fig. S9 and the results now read:

“ The scoring of mitochondrial ultrastructure on transmission electron micrographs showed that SKNMM^{MYCN} cells in the presence of doxycycline had more disrupted mitochondria on Day 4 and subsequent timepoints than did SKNMM^{MYCN} cells maintained in culture without doxycycline or the other cell lines (Figs. 4L-O, S9 and data not shown). Similar results were obtained for the ATRX-deficient U2OS^{MYCN} cells but not ATRX-wild type SKNBE2^{MYCN} cells (Fig. S9). “

6. In Figure 5, the authors demonstrate evidence of DDR in SKNMMmycn cells; again – is this cell lines restricted. As the authors propose a baseline replicative stress in ATRX mutant cells, it would have been informative to quantify the extent of DDR across the various NB cell lines with and without endogenous or engineered ATRX mutations and MYCN amp/increased expression. This will help strengthen their conclusion of synthetic lethality.

We thank the reviewer for this suggestion. We have analyzed γ H2AX in a panel of 11 different neuroblastoma cell lines including:

MYCN-amplified: SKNBE2, IMR32, NB-5 and NB1691

MYCN low copy number gain: NBL5

MYC high: SKNAS, SH-SY5Y

ATRX mutant: SKNMM, CHLA90

There was no correlation between γ H2AX level and the status of MYCN and/or ATRX. However, our hypothesis is that elevated MYCN or ATRX alone can lead to replicative stress and it is the combination of the two that is synthetic lethal. Indeed, we have a manuscript currently in review that shows neuroblastomas are unique among pediatric solid tumors in that they have a genome signature of ROS induced DNA damage. We are happy to provide a copy of that manuscript for the reviewers upon request.

7. The authors performed extensive epigenomic analyses and 2 drug screens to map down stream

effectors of MYCN/ATRX synergy. While the identification of retinoids and CUX2 is interesting, the significance and implications for understanding MYCN/ATRX synergy from these studies is not clearly articulated.

We have modified the discussion which now reads:

“ Here we show that amplification of the MYCN oncogene and inactivation of the ATRX tumor-suppressor gene are mutually exclusive in neuroblastomas from patients of all ages and stages of disease. One small, discrepant tumor sample may have contained 2 separate clones, but more detailed analysis was not possible due to limited tissue. In mouse models and cell lines, the combination of elevated MYCN expression and ATRX loss led to synthetic lethality. Ectopic MYCN caused metabolic reprogramming, mitochondrial dysfunction, ROS production, and DNA damage in ATRX-mutant neuroblastoma cells. We propose that MYCN amplification and ATRX mutations are incompatible in neuroblastoma, because both lead to DNA-replicative stress^{10,38,39}. Consistent with this model, the synthetic lethality was partially rescued by genes that reduce oxidative stress (CUX2) and pharmacological agents that induce differentiation (retinoic acid) or reduce ROS levels (N-acetyl cysteine). Similarly, pharmacological agents that induced replicative stress through DNA damage exacerbated the synthetic lethality. Based on data presented here, this synthetic lethality may extend to other MYC-driven tumors.”

Overall, I found this to be an interesting manuscript with an innovative finding. The authors include substantial amounts of detailed data - generally supportive of their premise, however, the rationale for execution and inclusion of specific experiments especially the drug screen was not clear. The authors may want to exploit their substantial data to better clarify more direct mechanisms to support their observation

Reviewer #3 (Remarks to the Author):

This study “MYCN amplification and ATRX mutations are incompatible in neuroblastoma” by Zeineldin et al reports the mutually exclusive relationship between amplification of the oncogene MYCN and mutation of the tumor suppressor ATRX in human neuroblastoma, and investigates the plausible molecular mechanism leading to it. They find that MYCN amplification causes metabolic reprogramming including enhanced Warburg effect, increased uptake of both glucose and glutamine, dysfunctional mitochondria and increased oxidative stress in the ATRX mutant cells. In general, the conclusions reached by the authors from the metabolic profiling experiments are supported by the current data. However, the presentation of the results could be improved. Specific comments are listed below.

1. Since M+4 is the most abundant mass isotopomer of aspartate, it would be helpful to also include it in the diagram in Figure 4A.

We have changed Fig. 4A.

2. It is not clear to me why only changes in M+5 glutamate and M+5 alpha-ketoglutarate are emphasized by the arrows in Figure 4B. Is that because all other changes are not statistically significant? If so, this needs to be clarified in the figure legend.

We apologize for the confusion. We emphasized the M+5 glutamate and alpha ketoglutarate because they are the most abundant and they are the direct derivatives of the labeled glutamine. We have the quantification of all forms of all the metabolites in Table S8. The legend for Figure 4B now reads:

“B) Bar plot of ¹³C-labeled isotopomers in SKNMM^{MYCN} cells after 4 days in the presence or absence of doxycycline and 5 hours of labeling with uniformly labeled ¹³C₅-glutamine. The arrows indicate an increase in M+5 glutamate and M+5 α -ketoglutarate (the most abundant and direct derivatives of ¹³C₅-glutamine) in the presence of doxycycline. Each bar is the mean and standard error of the mean of technical replicates.”

Minor:

Page 13: ‘Cells also upregulate pathways required to mitigate the reactive oxygen species (ROS) that are a natural biproduct of mitochondrial metabolism to prevent excessive protein or DNA damage’: ‘biproduct’ should be ‘byproduct’.

We have corrected this typo.

References

- 1 Leung, J. W., Ghosal, G., Wang, W., Shen, X., Wang, J., Li, L. & Chen, J. Alpha thalassaemia/mental retardation syndrome X-linked gene product ATRX is required for proper replication restart and cellular resistance to replication stress. *J Biol Chem* 288, 6342-6350, doi:10.1074/jbc.M112.411603 (2013).
- 2 Huh, M. S., Price O’Dea, T., Ouazia, D., McKay, B. C., Parise, G., Parks, R. J., Rudnicki, M. A. & Picketts, D. J. Compromised genomic integrity impedes muscle growth after Atrx inactivation. *J Clin Invest* 122, 4412-4423, doi:10.1172/jci63765 (2012).
- 3 Watson, L. A., Solomon, L. A., Li, J. R., Jiang, Y., Edwards, M., Shin-ya, K., Beier, F. & Berube, N. G. Atrx deficiency induces telomere dysfunction, endocrine defects, and reduced life span. *J Clin Invest* 123, 2049-2063, doi:10.1172/jci65634 (2013).
- 4 Huh, M. S., Ivanochko, D., Hashem, L. E., Curtin, M., Delorme, M., Goodall, E., Yan, K. & Picketts, D. J. Stalled replication forks within heterochromatin require ATRX for protection. *Cell death & disease* 7, e2220, doi:10.1038/cddis.2016.121 (2016).
- 5 Dyer, M. A., Qadeer, Z. A., Valle-Garcia, D. & Bernstein, E. ATRX and DAXX: Mechanisms and Mutations. *Cold Spring Harb Perspect Med* 7, doi:10.1101/cshperspect.a026567 (2017).
- 6 Clynes, D., Higgs, D. R. & Gibbons, R. J. The chromatin remodeler ATRX: a repeat offender in human disease. *Trends in biochemical sciences* 38, 461-466, doi:10.1016/j.tibs.2013.06.011 (2013).
- 7 Wang, Y., Yang, J., Wild, A. T., Wu, W. H., Shah, R., Danussi, C., Riggins, G. J., Kannan, K., Sulman, E. P., Chan, T. A. & Huse, J. T. G-quadruplex DNA drives genomic instability and represents a targetable molecular abnormality in ATRX-deficient malignant glioma. *Nat Commun* 10, 943, doi:10.1038/s41467-019-08905-8 (2019).
- 8 Prorok, P., Artufel, M., Aze, A., Coulombe, P., Peiffer, I., Lacroix, L., Guedin, A., Mergny, J. L., Damaschke, J., Schepers, A., Ballester, B. & Mechali, M. Involvement of G-quadruplex regions in mammalian replication origin activity. *Nat Commun* 10, 3274, doi:10.1038/s41467-019-11104-0 (2019).
- 9 Zyner, K. G., Mulhearn, D. S., Adhikari, S., Martinez Cuesta, S., Di Antonio, M., Erard, N., Hannon, G. J., Tannahill, D. & Balasubramanian, S. Genetic interactions of G-quadruplexes in humans. *eLife* 8, doi:10.7554/eLife.46793 (2019).
- 10 Skene, P. J. & Henikoff, S. An efficient targeted nuclease strategy for high-resolution mapping of DNA binding sites. *eLife* 6, doi:10.7554/eLife.21856 (2017).
- 11 Skene, P. J., Henikoff, J. G. & Henikoff, S. Targeted in situ genome-wide profiling with high efficiency for low cell numbers. *Nat Protoc* 13, 1006-1019, doi:10.1038/nprot.2018.015 (2018).
- 12 Meers, M. P., Bryson, T. D., Henikoff, J. G. & Henikoff, S. Improved CUT&RUN chromatin profiling tools. *eLife* 8, doi:10.7554/eLife.46314 (2019).

- 13 Millau, J. F. & Gaudreau, L. CTCF, cohesin, and histone variants: connecting the genome. *Biochemistry and cell biology = Biochimie et biologie cellulaire* 89, 505-513, doi:10.1139/o11-052 (2011).
- 14 Weth, O., Paprotka, C., Gunther, K., Schulte, A., Baierl, M., Leers, J., Galjart, N. & Renkawitz, R. CTCF induces histone variant incorporation, erases the H3K27me3 histone mark and opens chromatin. *Nucleic Acids Res* 42, 11941-11951, doi:10.1093/nar/gku937 (2014).
- 15 Jin, C., Zang, C., Wei, G., Cui, K., Peng, W., Zhao, K. & Felsenfeld, G. H3.3/H2A.Z double variant-containing nucleosomes mark 'nucleosome-free regions' of active promoters and other regulatory regions. *Nature genetics* 41, 941-945, doi:10.1038/ng.409 (2009).
- 16 Wang, J. Y., Lin, C. H., Yang, C. H., Tan, T. H. & Chen, Y. R. Biochemical and biological characterization of a neuroendocrine-associated phosphatase. *J Neurochem* 98, 89-101, doi:10.1111/j.1471-4159.2006.03852.x (2006).
- 17 Wang, J. Y., Yang, C. H., Yeh, C. L., Lin, C. H. & Chen, Y. R. NEAP causes down-regulation of EGFR, subsequently induces the suppression of NGF-induced differentiation in PC12 cells. *J Neurochem* 107, 1544-1555, doi:10.1111/j.1471-4159.2008.05714.x (2008).
- 18 Shang, X., Vasudevan, S. A., Yu, Y., Ge, N., Ludwig, A. D., Wesson, C. L., Wang, K., Burlingame, S. M., Zhao, Y. J., Rao, P. H., Lu, X., Russell, H. V., Okcu, M. F., Hicks, M. J., Shohet, J. M., Donehower, L. A., Nuchtern, J. G. & Yang, J. Dual-specificity phosphatase 26 is a novel p53 phosphatase and inhibits p53 tumor suppressor functions in human neuroblastoma. *Oncogene* 29, 4938-4946, doi:10.1038/onc.2010.244 (2010).
- 19 Nguyen, D. T., Voon, H. P. J., Xella, B., Scott, C., Clynes, D., Babbs, C., Ayyub, H., Kerry, J., Sharpe, J. A., Sloane-Stanley, J. A., Butler, S., Fisher, C. A., Gray, N. E., Jenuwein, T., Higgs, D. R. & Gibbons, R. J. The chromatin remodelling factor ATRX suppresses R-loops in transcribed telomeric repeats. *EMBO reports* 18, 914-928, doi:10.15252/embr.201643078 (2017).
- 20 Schwab, R. A., Nieminuszczy, J., Shah, F., Langton, J., Lopez Martinez, D., Liang, C. C., Cohn, M. A., Gibbons, R. J., Deans, A. J. & Niedzwiedz, W. The Fanconi Anemia Pathway Maintains Genome Stability by Coordinating Replication and Transcription. *Molecular cell* 60, 351-361, doi:10.1016/j.molcel.2015.09.012 (2015).
- 21 Arora, R., Lee, Y., Wischniewski, H., Brun, C. M., Schwarz, T. & Azzalin, C. M. RNaseH1 regulates TERRA-telomeric DNA hybrids and telomere maintenance in ALT tumour cells. *Nat Commun* 5, 5220, doi:10.1038/ncomms6220 (2014).
- 22 Flynn, R. L., Cox, K. E., Jeitany, M., Wakimoto, H., Bryll, A. R., Ganem, N. J., Bersani, F., Pineda, J. R., Suva, M. L., Benes, C. H., Haber, D. A., Boussin, F. D. & Zou, L. Alternative lengthening of telomeres renders cancer cells hypersensitive to ATR inhibitors. *Science* 347, 273-277, doi:10.1126/science.1257216 (2015).
- 23 Althoff, K., Beckers, A., Bell, E., Nortmeyer, M., Thor, T., Sprussel, A., Lindner, S., De Preter, K., Florin, A., Heukamp, L. C., Klein-Hitpass, L., Astrahantseff, K., Kumps, C., Speleman, F., Eggert, A., Westermann, F., Schramm, A. & Schulte, J. H. A Cre-conditional MYCN-driven neuroblastoma mouse model as an improved tool for preclinical studies. *Oncogene* 34, 3357-3368, doi:10.1038/onc.2014.269 (2015).
- 24 Berry, T., Luther, W., Bhatnagar, N., Jamin, Y., Poon, E., Sanda, T., Pei, D., Sharma, B., Vetharoy, W. R., Hallsworth, A., Ahmad, Z., Barker, K., Moreau, L., Webber, H., Wang, W., Liu, Q., Perez-Atayde, A., Rodig, S., Cheung, N. K., Raynaud, F., Hallberg, B., Robinson, S. P., Gray, N. S., Pearson, A. D., Eccles, S. A., Chesler, L. & George, R. E. The ALK(F1174L) mutation potentiates the oncogenic activity of MYCN in neuroblastoma. *Cancer Cell* 22, 117-130, doi:10.1016/j.ccr.2012.06.001 (2012).
- 25 Berube, N. G., Mangelsdorf, M., Jagla, M., Vanderluit, J., Garrick, D., Gibbons, R. J., Higgs, D. R., Slack, R. S. & Picketts, D. J. The chromatin-remodeling protein ATRX is critical for neuronal survival during corticogenesis. *The Journal of clinical investigation* 115, 258-267, doi:10.1172/JCI22329 (2005).

- 26 Bagheri-Fam, S., Argentaro, A., Svingen, T., Combes, A. N., Sinclair, A. H., Koopman, P. & Harley, V. R. Defective survival of proliferating Sertoli cells and androgen receptor function in a mouse model of the ATR-X syndrome. *Human molecular genetics* 20, 2213-2224, doi:10.1093/hmg/ddr109 (2011).
- 27 Downing, J. R., Wilson, R. K., Zhang, J., Mardis, E. R., Pui, C. H., Ding, L., Ley, T. J. & Evans, W. E. The Pediatric Cancer Genome Project. *Nature genetics* 44, 619-622, doi:10.1038/ng.2287 (2012).
- 28 Cheung, N. K., Zhang, J., Lu, C., Parker, M., Bahrami, A., Tickoo, S. K., Heguy, A., Pappo, A. S., Federico, S., Dalton, J., Cheung, I. Y., Ding, L., Fulton, R., Wang, J., Chen, X., Becksfort, J., Wu, J., Billups, C. A., Ellison, D., Mardis, E. R., Wilson, R. K., Downing, J. R. & Dyer, M. A. Association of age at diagnosis and genetic mutations in patients with neuroblastoma. *Jama* 307, 1062-1071, doi:10.1001/jama.2012.228 (2012).
- 29 De Wilde, B., Beckers, A., Lindner, S., Kristina, A., De Preter, K., Depuydt, P., Mestdagh, P., Sante, T., Lefever, S., Hertwig, F., Peng, Z., Shi, L. M., Lee, S., Vandermarliere, E., Martens, L., Menten, B., Schramm, A., Fischer, M., Schulte, J., Vandesompele, J. & Speleman, F. The mutational landscape of MYCN, Lin28b and ALK(F1174L) driven murine neuroblastoma mimics human disease. *Oncotarget* 9, 8334-8349, doi:10.18632/oncotarget.23614 (2018).

REVIEWERS' COMMENTS:

Reviewer #1 (Remarks to the Author):

We appreciate the careful reply to our comments and where necessary the additional experiments performed, which in our opinion add further to the significance of this manuscript.

Reviewer #2 (Remarks to the Author):

The authors have thoroughly addressed all my concerns - the manuscript is significantly improved

Reviewer #3 (Remarks to the Author):

I am satisfied.